# Stable Offline Value Function Learning with Bisimulation-based Representations

**Brahma S. Pavse** [1]  **Yudong Chen** [1]  **Qiaomin Xie** [1]  **Josiah P. Hanna** [1]

## Abstract

In reinforcement learning, offline value function learning is the procedure of using an offline dataset to estimate the expected discounted return from each state when taking actions according to a fixed target policy. The stability of this procedure, i.e., whether it converges to its fixed-point, critically depends on the representations of the state-action pairs. Poorly learned representations can make value function learning unstable, or even divergent. Therefore, it is critical to stabilize value function learning by explicitly shaping the state-action representations. Recently, the class of bisimulation-based algorithms have shown promise in shaping representations for control. However, it is still unclear if this class of methods can *stabilize* value function learning. In this work, we investigate this question and answer it affirmatively. We introduce a bisimulation-based algorithm called kernel representations for offline policy evaluation (KROPE). KROPE uses a kernel to shape state-action representations such that state-action pairs that have similar immediate rewards and lead to similar next state-action pairs under the target policy also have similar representations. We show that KROPE: 1) learns stable representations and 2) leads to lower value error than baselines. Our analysis provides new theoretical insight into the stability properties of bisimulation-based methods and suggests that practitioners can use these methods to improve the stability and accuracy of offline evaluation of reinforcement learning agents.

## 1. Introduction

Learning the value function of a policy is a critical component of many reinforcement learning (RL) algorithms

($Sutton$ & $Barto$, 2018). While value function learning algorithms such as temporal-difference learning (TD) have been successful, they can be unreliable. In particular, the deadly triad, i.e., the combination of off-policy updates, function approximation, and bootstrapping, can make TD-based methods diverge (Sutton & Barto, 2018; Tsitsiklis & Van Roy, 1997; Baird, 1995; Hasselt et al., 2018). Function approximation is a critical component of value function learning since it determines the representations of state-action pairs, which in turn defines the space of expressible value functions. Depending on how this value function space is represented, value function learning algorithms may diverge (Ghosh & Bellemare, 2020). That is, the value function learning algorithm may not converge to its fixed-point, or may even diverge away from it. In this work, we explicitly learn state-action representations to improve the stability of offline value function learning.

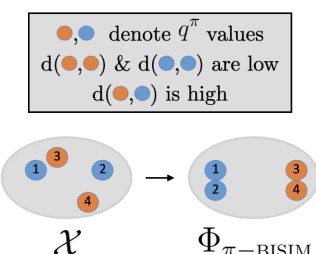

*Figure 1.* The figure illustrates the native state-action representations $\mathcal{X}$ and $\pi$-bisimulation representations $\Phi_{\pi-\mathrm{BISIM}}$. $\pi$-bisimulation algorithms use a distance function $d$ that captures differences between state-action pairs based on immediate rewards and differences of next state-action pairs under $\pi$ to shape their representations. Ultimately, the goal of $\pi$-bisimulation methods is to learn representations such that state-actions pairs with similar action-values under $\pi$ also have similar representations. The function $d$ outputs low values within the blue (and orange) state-actions but high values between blue and orange state-actions. Therefore, the blue (and orange) state-actions have similar representations, but different representations between the distinct colors.

In seeking such representations, we turn to $\pi$-bisimulation algorithms. These algorithms define a metric to capture *behavioral similarity* between state-action pairs such that similarity is based on immediate rewards received and the similarity of next state-action pairs visited by $\pi$ (Castro,

[1]University of Wisconsin – Madison. Correspondence to: Brahma S. Pavse <pavse@wisc.edu>.

2020). The algorithms then use this metric to learn representations such that state-action pairs that are similar under this metric have similar representations. Ultimately, the goal of $\pi$-bisimulation methods is to learn representations such that state-actions pairs with similar values under $\pi$ also have similar representations (see Figure 1). While these algorithms have shown promise in improving the expected return of RL algorithms, it remains unclear whether they contribute to stability (Castro et al., 2023; Zhang et al., 2021; Castro et al., 2021). In this paper, we aim to understand whether the $\pi$-bisimulation-based representations stabilize value function learning.

In this work, we focus on offline value function learning. Given a fixed, offline dataset generated by unknown and possibly multiple behavior policies, the goal is to estimate the value function of a fixed, target policy. Towards establishing the stability properties of $\pi$-bisimulation representations, we introduce kernel representations for offline policy evaluation (KROPE). KROPE defines a kernel that captures similarity between state-action pairs based on immediate rewards received and similarity of next state-action pairs under the *target* policy. It then shapes the state-action representations such that state-action pairs that are similar according to this kernel have similar representations. We use KROPE as the representative algorithm for the class of bisimulation-based representation learning algorithms to investigate the following question:

***Can bisimulation-based representation learning stabilize offline value function learning?***

Through theoretical and empirical analysis, we answer this question affirmatively and make the following contributions:

1. We introduce kernel representations for offline policy evaluation (KROPE) for stable and accurate offline value function learning (Section 3).

2. We prove that KROPE's representations stabilize least-squares policy evaluation (LSPE), a popular value function learning algorithm (Sections 3.2).

3. We prove that KROPE representations are Bellman complete, another indication of stability (Sections 3.3).

4. We empirically validate that KROPE representations improve the stability and accuracy of offline value function learning algorithms on $10/13$ offline datasets and against 7 baselines (Section 4).

5. We empirically analyze the sensitivity of the KROPE learning procedure under the deadly triad. These experiments shed light on when representation learning may be easier than value function learning (Section 4.4).

## 2. Background

In this section, we present our problem setup and discuss prior work.

### 2.1. Problem Setup and Notation

We consider the infinite-horizon Markov decision process (MDP) framework (Puterman, 2014), $\mathcal{M} = \langle \mathcal{S}, \mathcal{A}, r, P, \gamma, d_0 \rangle$, where $\mathcal{S}$ is the state space, $\mathcal{A}$ is the action space, $r : \mathcal{S} \times \mathcal{A} \to [-1, 1]$ is the deterministic reward function, $P : \mathcal{S} \times \mathcal{A} \to \Delta(\mathcal{S})$ is the transition dynamics function, $\gamma \in [0, 1)$ is the discount factor, and $d_0 \in \Delta(\mathcal{S})$ is the initial state distribution, where $\Delta(X)$ represents the set of all probability distributions over a set $X$. We refer to the joint state-action space as $\mathcal{X} := \mathcal{S} \times \mathcal{A}$. The agent acting according to policy $\pi : \mathcal{S} \to \Delta(\mathcal{A})$ in the MDP generates a trajectory: $S_0, A_0, R_0, S_1, A_1, R_1, ...$, where $S_0 \sim d_0$, $A_t \sim \pi(\cdot|S_t)$, $R_t := r(S_t, A_t)$, and $S_{t+1} \sim P(\cdot|S_t, A_t)$ for $t \geq 0$.

We define the action-value function of a policy $\pi$ for a given state-action pair as $q^\pi(s, a) := \mathbb{E}_\pi[\sum_{t=0}^\infty \gamma^t r(S_t, A_t)|S_0 = s, A_0 = a]$, i.e., the expected discounted return when starting from state $s$ with initial action $a$ and then following policy $\pi$. The Bellman evaluation operator $\mathcal{T}^\pi : \mathbb{R}^\mathcal{X} \to \mathbb{R}^\mathcal{X}$ is defined as $(\mathcal{T}^\pi f)(s, a) := r(s, a) + \gamma \mathbb{E}_{S' \sim P(\cdot|s,a), A' \sim \pi}[f(S', A')], \forall f \in \mathbb{R}^\mathcal{X}$. Accordingly, the action-value function satisfies the Bellman equation, i.e., $q(s, a) = r(s, a) + \gamma \mathbb{E}_{P, \pi_e}[q(S', A')]$.

It will be convenient to consider the matrix notation equivalents of the above functions. Since a policy $\pi$ induces a Markov chain on $\mathcal{X}$, we can denote the transition matrix of this Markov chain by $P^\pi \in \mathbb{R}^{|\mathcal{X}| \times |\mathcal{X}|}$. Here, each entry $P^\pi(i, j)$ is the probability of transitioning from state-actions $i$ to $j$. Similarly, we have the action-value function $q^\pi \in \mathbb{R}^{|\mathcal{X}|}$ and reward vector $r \in \mathbb{R}^{|\mathcal{X}|}$, where the entry $q^\pi(i)$ and $r(i)$ are the expected discounted return from state-action $i$ under $\pi$ and reward received at state-action $i$ respectively.

In this work, we study the representations of the state-action space. We use $\phi : \mathcal{S} \times \mathcal{A} \to \mathbb{R}^d$ to denote the state-action representations, which maps state-action pairs into a $d$-dimensional Euclidean space. We denote the matrix of all the state-action features as $\Phi \in \mathbb{R}^{|\mathcal{X}| \times d}$, where each row is the state-action feature $\phi(s, a) \in \mathbb{R}^d$ for state-action pair $(s, a)$. When dealing with the offline dataset $\mathcal{D}$, $\Phi$'s dimensions are $|\mathcal{D}| \times d$, where $|\mathcal{D}|$ is the number state-actions in the dataset $\mathcal{D}$. Note that $\Phi$ can be the native state-action features of the MDP, or the output of some representation learning algorithm, or the penultimate features of the action-value function when using a neural network. Throughout this paper, we will view $\phi$ as an encoder or state-action abstraction (Li et al., 2006). Note that the state-

action abstraction view enables us to view $\phi$ as a state-action aggregator from the space of state-actions $\mathcal{X}$ to the space of state-action groups $\mathcal{X}^\phi$.

## 2.2. Offline Policy Evaluation and Value Function Learning

In offline policy evaluation (OPE), the goal is to evaluate a fixed target policy, $\pi_e$, using a fixed dataset of $m$ transition tuples $\mathcal{D} := \{(s_i, a_i, s_i', r_i)\}_{i=1}^m$. In this work, we evaluate $\pi_e$ by estimating the *action-value function* $q^{\pi_e}$ using $\mathcal{D}$. Crucially, $\mathcal{D}$ may have been generated by a set of *unknown behavior* policies that are different from $\pi_e$, which means that simply averaging the discounted returns in $\mathcal{D}$ will produce an inconsistent estimate of $q^{\pi_e}$. In our theoretical results, we make the standard coverage assumption that $\forall s \in \mathcal{S}, \forall a \in \mathcal{A}$ if $\pi_e(a|s) > 0$, then the state-action pair $(s, a)$ has non-zero probability of appearing in $\mathcal{D}$ (Sutton & Barto, 2018; Precup et al., 2000).

We measure the accuracy of the value function estimate with the *mean squared value error* (MSVE). Let $\hat{q}^{\pi_e}$ be the estimate returned by a value function learning method using $\mathcal{D}$. The MSVE of this estimate is defined as $\text{MSVE}[\hat{q}^{\pi_e}] := \mathbb{E}_{(S,A)\sim\mathcal{D}}[(\hat{q}^{\pi_e}(S, A) - q^{\pi_e}(S, A))^2]$. In environments with continuous state-action spaces, where it is impossible to compute $q^{\pi_e}$ for all state-actions, we adopt a common evaluation procedure from the OPE literature of measuring the MSE across only the initial state-action distribution, i.e., $\text{MSE}[\hat{q}^{\pi_e}] := \mathbb{E}_{S_0\sim d_0, A_0\sim\pi_e}[(\hat{q}^{\pi_e}(S_0, A_0) - q^{\pi_e}(S_0, A_0))^2]$. For this procedure, we assume access to $d_0$ (Voloshin et al., 2021; Fu et al., 2021). While in practice $q^{\pi_e}$ is unknown, it is standard for the sake of empirical analysis to estimate $q^{\pi_e}$ by executing unbiased Monte Carlo rollouts of $\pi_e$ or computing $q^{\pi_e}$ exactly using dynamic programming in tabular environments (Voloshin et al., 2021; Fu et al., 2021).

**Least-Squares Policy Evaluation**   Least-squares policy evaluation (LSPE) is a value function learning algorithm, which models the action-value function as a linear function: $\hat{q}_\theta^{\pi_e}(s, a) := \phi(s, a)^\top \theta$, where $\theta \in \mathbb{R}^d$ (Nedic & Bertsekas, 2003). LSPE iteratively learns $\theta$ with the following updates per iteration step $t$:

$$\theta_{t+1} \leftarrow (\mathbb{E}_\mathcal{D}[\Phi^\top \Phi])^{-1} \mathbb{E}_{\mathcal{D},\pi_e}[\Phi^\top(r + \gamma P^{\pi_e}\Phi\theta_t)], \quad (1)$$

where the expectations are taken with respect to the randomness of the dataset $\mathcal{D}$ and $\pi_e$. Note that $\mathbb{E}[\Phi^\top\Phi]$ is the feature covariance matrix. Assuming LSPE converges, it will converge to the same fixed-point as TD(0) (Szepesvari, 2010), which we denote as $\theta_{\text{LSPE}}$. In this work, we follow a two-stage approach to applying LSPE: we first obtain the encoder $\phi$ either through representation learning or using the native features of the MDP, and then feed the obtained $\phi$ along with $\mathcal{D}$ and $\pi_e$ as input to LSPE, which outputs $\hat{q}_\theta^{\pi_e}$ (Nedic & Bertsekas, 2003; Chang et al., 2022). This

two-stage approach of learning a linear function on top of fixed representations is called the linear evaluation protocol (Chang et al., 2022; Farebrother et al., 2023; 2024; Grill et al., 2020; He et al., 2020). This protocol enables us to cleanly analyze the nature of the learned representations within the context of well-understood value function learning algorithms such as LSPE. In Appendix A, we include the pseudo-code for LSPE.

## 2.3. Stable Representations and $q^{\pi_e}$-Consistency

We define stability of LSPE and related TD-methods following (Ghosh & Bellemare, 2020):

**Definition 1** (Stability). LSPE *is said to be stable if for any initial* $\theta_0 \in \mathbb{R}^d$, $\lim_{t\to\infty}\theta_t = \theta_{\text{LSPE}}$ *when $\theta_t$ is updated according to Equation (1).*

When determining the stability of LSPE, we have the following proposition from prior work:

**Proposition 1** (Asadi et al. 2024; Wang et al. 2021). LSPE *is stable if and only if the spectral radius of* $(\mathbb{E}[\Phi^\top\Phi])^{-1}(\gamma\mathbb{E}[\Phi^\top P^{\pi_e}\Phi])$, *i.e., its maximum absolute eigenvalue, is less than* 1.

Therefore, the stability of LSPE largely depends on the representations $\Phi$ and the distribution shift between the data distribution of $\mathcal{D}$ and $\pi_e$. In this work, we study the stability of LSPE for a fixed distribution of $\mathcal{D}$ and learn $\Phi$. If a given $\Phi$ stabilizes LSPE, we say $\Phi$ is a *stable representation*.

In addition to stability, we also want the state-action features to be such that state-action features that are close in the representation space also have similar $q^{\pi_e}$ values (Lyle et al., 2022; Lan et al., 2021). In this work, we call this property $q^{\pi_e}$-*consistency* since the learned representations are consistent with the $q^{\pi_e}$ values.

## 2.4. Related Works

In this section, we discuss the most relevant prior literature on OPE and representation learning.

**Representations for Offline RL and OPE.**   There are several works that have shown shaping representations can be effective for offline RL (Yang & Nachum, 2021; Islam et al., 2023; Nachum & Yang, 2024; Zang et al., 2023a; Arora et al., 2020; Uehara et al., 2022; Chen & Jiang, 2019; Pavse & Hanna, 2023b). Ghosh & Bellemare (2020) presented a theoretical understanding of how various representations can stabilize TD learning. However, they did not discuss bisimulation-based representations. Kumar et al. (2022); Ma et al. (2024); He et al. (2024) promote the stability of TD-based methods by increasing the rank of the representations to prevent representation collapse. However, as we show in Section 4, these types of representations can still lead to inaccurate OPE. On the other hand, KROPE mit-

igates representation collapse and leads to accurate OPE. Chang et al. (2022) introduced BCRL to learn Bellman complete representations for stable OPE. While in theory, BC representations are desirable, we found that BCRL is sensitive to hyperparameter tuning. In contrast, we show that KROPE is more robust to hyperparameter tuning. Pavse & Hanna (2023a) showed that bisimulation-based representations mitigate the divergence of FQE; however, they did not provide an explanation for divergence mitigation. Our work provides theoretical insight into the stability properties of bisimulation-based algorithms.

**Bisimulation-based Representation Learning.** Recently, there has been lot of interest in $\pi$-bisimulation algorithms for better generalization (Ferns et al., 2004; 2011; Ferns & Precup, 2014; Castro, 2020; Zang et al., 2023b). These algorithms measure similarity between two state-action pairs based on immediate rewards received and the similarity of next state-action pairs visited by $\pi$. These algorithms first define a distance metric that captures this $\pi$-bisimilarity, and then use this metric to learn representations such that $\pi$-bisimular states have similar representations (Castro et al., 2021; Castro, 2020; Zhang et al., 2021; Castro et al., 2023; Chen & Pan, 2022; Kemertas & Jepson, 2022). Castro (2020) introduced a $\pi$-bisimulation learning algorithm but assumed that the transition dynamics are deterministic. Zhang et al. (2021) introduced a bisimulation algorithm that allowed for Gaussian dynamics and demonstrated its effectiveness in discarding distracting features for control. Castro et al. (2021; 2023) introduced bisimulation-based algorithms for control that allow for stochastic transition dynamics. To the best of our knowledge, there is still a gap in the literature as to whether the general class of these learned $\pi$-bisimulation-based representations stabilize offline value function learning. To address this gap, we introduce KROPE, an algorithm that leverages Castro et al. (2023)'s kernel perspective on similarity metrics. We first explicitly define a kernel that captures the relationship between state-action features in latent space in terms of the $\pi$-bisimulation relation. This definition allows us to immediately establish stability since we can now analyze the spectral properties of offline value function learning algorithms. The proofs for KROPE's basic theoretical properties (Section 3.1) follow those by Castro et al. (2023). Our stability-related theoretical results (Sections 3.2 and 3.3) and empirical analysis (Section 4) are novel to this work.

## 3. Kernel Representations for Offline Policy Evaluation

We now present our bisimulation-based representation learning algorithm, kernel representations for OPE (KROPE). We present the desired KROPE kernel, prove stability properties of KROPE representations, and present a practical learning

algorithm to learn them. We defer the proofs to Appendix B.

### 3.1. KROPE Kernel

Prior works typically define a *distance* metric to capture the notion of $\pi$-bisimilarity between two states: the distance between states is in terms of the immediate rewards received and differences of next states under $\pi$ (see Figure 1) (Castro, 2020). In this work, we follow Castro et al. (2023) and take a perspective of kernels instead of distances. We first define a kernel that captures $\pi$-bisimilarity $k^{\pi_e} : \mathcal{X} \times \mathcal{X} \to \mathbb{R}$, but for pairs of state-actions and under $\pi_e$:

$$k^{\pi_e}(s_1, a_1; s_2, a_2) = k_1(s_1, a_1; s_2, a_2)$$
$$+ \gamma k_2(k^{\pi_e})(P^{\pi_e}(\cdot|s_1, a_1), P^{\pi_e}(\cdot|s_2, a_2)). \quad (2)$$

where $k_1(s_1, a_1; s_2, a_2) := 1 - \frac{|r(s_1, a_1) - r(s_2, a_2)|}{|r_{\max} - r_{\min}|}$ and $k_2(k^{\pi_e})(P^{\pi_e}(\cdot|s_1, a_1), P^{\pi_e}(\cdot|s_2, a_2)) :=$ $\mathbb{E}_{s_1', a_1' \sim P^{\pi_e}(\cdot|s_1, a_1), s_2', a_2' \sim P^{\pi_e}(\cdot|s_2, a_2)}[k^{\pi_e}(s_1', a_1'; s_2', a_2')]$. Here, $k_1$ measures short-term similarity based on rewards received and $k_2$ measures long-term similarity between probability distributions by measuring similarity between samples of the distributions according to $k^{\pi_e}$ (Castro et al., 2023). We refer to $k^{\pi_e}$ as the KROPE kernel.

**Remarks on $k^{\pi_e}$.** We now discuss the trade-offs of $k^{\pi_e}$. While $k^{\pi_e}$ is different from prior work, it benefits and suffers from the same advantages and disadvantages discussed in Castro et al. (2023).

By definition $k_2$ is a function of independently coupling $(s_1', a_1')$ and $(s_2', a_2')$ (Castro et al., 2021; 2023). The advantage of this independence is that it enables a practical algorithm and is flexible as it make no assumptions on the nature of the transition dynamics of $P^{\pi_e}$. While independent coupling does raise a complication with stochastic dynamics and/or stochastic policies since self-similarity may be lower than similarity between different state-actions, i.e., $k^{\pi_e}(s_1, a_1; s_1, a_1) < k^{\pi_e}(s_1, a_1; s_2, a_2)$ for some $(s_1, a_1), (s_2, a_2) \in \mathcal{X}$, $k^{\pi_e}$ still induces a valid $\pi$-bisimulation-based metric. To see its validity, consider the induced distance function by $k^{\pi_e}$, i.e., $\forall x, y \in \mathcal{X}, d_{\text{KROPE}}(x, y) := k^{\pi_e}(x, x) + k^{\pi_e}(y, y) - 2k^{\pi_e}(x, y)$. The metric $d_{\text{KROPE}}$ satisfies continuity (see Lemma 3 in Appendix B.1), which is an important property for metrics to satisfy (Lan et al., 2021). Lemma 3 states that the absolute action-value difference between any two state-action pairs under $\pi_e$ is upper-bounded by $d_{\text{KROPE}}$ plus an additive constant, where the additive constant arises due to the independent coupling. Thus far, prior work has removed the additive constant by assuming deterministic or Gaussian dynamics (Castro, 2020; Zhang et al., 2021). However, these assumptions are practically more restrictive. While one might expect the existence of the additive constant to hurt performance, prior work and our work (see Section 4)

show that independent coupling actually yields strong performance (Pavse & Hanna, 2023a; Castro et al., 2023). Furthermore, in Section 3.2 we show that this property does not affect the stability properties of the learned representations. An interesting future direction will be to develop practically feasible approaches to eliminate the additive constant.

Since Lemma 3 and the associated contraction, metric space completeness, and fixed-point uniqueness properties are similar to Castro et al. (2023)'s kernel, we defer these results to Appendix B.1.

## 3.2. Stability of KROPE Representations

In the previous section, we defined the KROPE kernel. Ultimately, however, we are interested in *representations* that satisfy the relationship in Equation (2). We modify Equation (2) accordingly by giving $k^{\pi_e}$ some functional form in terms of state-action representations. We do so with the dot product: $\langle u, v \rangle = u^\top v, \forall u, v \in \mathbb{R}^d$, i.e., $k^{\pi_e}(s_1, a_1; s_2, a_2) = \phi(s_1, a_1)^\top \phi(s_2, a_2)$. With this setup, we write Equation (2) in matrix notation and define the KROPE representations as follows:

**Definition 2** (KROPE Representations). *Consider state-action representations $\Phi \in \mathbb{R}^{|\mathcal{X}| \times d}$ that are embedded in $\mathbb{R}^d$ with $k^{\pi_e}(s_1, a_1; s_2, a_2) = \phi(s_1, a_1)^\top \phi(s_2, a_2)$. We say $\Phi$ is a KROPE representation if it satisfies the following:*

$$\mathbb{E}_\mathcal{D}[\Phi\Phi^\top] = \mathbb{E}_\mathcal{D}[K_1] + \gamma\mathbb{E}_{\mathcal{D},\pi_e}[P^{\pi_e}\Phi(P^{\pi_e}\Phi)^\top] \quad (3)$$

*where each entry of $K_1 \in \mathbb{R}^{|\mathcal{X}| \times |\mathcal{X}|}$ represents the short-term similarity, $k_1$, between every pair of state-actions, i.e., $K_1(s_1, a_1; s_2, a_2) := 1 - \frac{|r(s_1, a_1) - r(s_2, a_2)|}{|r_{max} - r_{min}|}$.*

Given this definition, we present our novel result proving the stability of KROPE representations:

> **Theorem 1.** *If $\Phi$ is a KROPE representation as defined in Definition 2, then the spectral radius of $(\mathbb{E}[\Phi^\top\Phi]))^{-1}\mathbb{E}[\gamma\Phi^\top P^{\pi_e}\Phi]$ is less than 1. That is, $\Phi$ stabilizes LSPE.*

By adopting the kernel perspective, Theorem 1, proved in Appendix B.2, tells us that $\pi$-bisimulation-based KROPE representations stabilize OPE with LSPE. Intuitively, they are stable since when $(\mathbb{E}[\Phi^\top\Phi]))^{-1}\mathbb{E}[\gamma\Phi^\top P^{\pi_e}\Phi]$'s spectral radius is less than 1, each update to $\theta_t$ in Equation (1) is non-expansive. That is, each update brings $\theta_t$ closer to $\theta_{\text{LSPE}}$.

## 3.3. Connection to Bellman Completeness

In this section, we draw a novel connection between KROPE representations and Bellman completeness. Chen & Jiang (2019) proved a similar result but focused on bisimulation

representations instead of KROPE representations, which are $\pi$-bisimulation-like representations. We say a function class $\mathcal{F}$ is Bellman complete if it is complete under the Bellman operator: $\mathcal{T}^{\pi_e} f \subseteq \mathcal{F}, \forall f \in \mathcal{F}$. For instance, suppose $\mathcal{F}$ is the class of linear functions spanned by $\Phi$, $\mathcal{F} := \{f \in \mathbb{R}^\mathcal{X} : f := \Phi w\}, w \in \mathbb{R}^d$. Then if $\mathcal{T}^{\pi_e} f, \forall f \in \mathcal{F}$ is also a linear function within the span of $\Phi$, we say $\Phi$ is a Bellman complete representation. Bellman completeness is an alternative condition for stability and is typically assumed to ensure to data-efficient policy evaluation (Wang et al., 2021; Szepesvári & Munos, 2005; Chang et al., 2022).

Recall, that the induced distance function due to $k^{\pi_e}$ is

$$\forall x, y \in \mathcal{X} : d_{\text{KROPE}}(x, y) := k^{\pi_e}(x, x) + k^{\pi_e}(y, y) - 2k^{\pi_e}(x, y).$$

We now present our second main result. It states that KROPE representations are Bellman complete under a reward function injectivity assumption (see proof in Appendix B.2):

> **Theorem 2.** *Let $\phi : \mathcal{X} \to \mathcal{X}^\phi$ be the state-action abstraction induced by grouping state-actions $x, y \in \mathcal{X}$ such that if $d_{\text{KROPE}}(x, y) = 0$, then $\phi(x) = \phi(y), \forall x, y \in \mathcal{X}$. Then $\phi$ is Bellman complete if the abstract reward function $r^\phi : \mathcal{X}^\phi \rightarrowtail (-1, 1)$ is injective (i.e., distinct abstract rewards).*

> **Takeaway #1: Stability of Bisimulation-based Representations**
>
> Under the theoretical assumptions made, KROPE representations avoid divergence of offline value function learning. They 1) induce non-expansive value function updates and 2) are Bellman complete.

## 3.4. KROPE Learning Algorithm

In this section, we present an algorithm that learns the KROPE representations from data. We include the pseudo-code of KROPE in Appendix A. The KROPE learning algorithm uses an encoder $\phi_\omega : \mathcal{S} \times \mathcal{A} \to \mathbb{R}^d$, which is parameterized by weights $\omega$ of a function approximator. It then parameterizes the kernel with the dot product, i.e, $\tilde{k}_\omega(s_1, a_1; s_2, a_2) := \phi_\omega(s_1, a_1)^\top \phi_\omega(s_2, a_2)$ (see Equation (3)). Finally, the algorithm then minimizes the following loss function, which is similar to how the value function is learned in deep RL (Mnih et al., 2015):

$$\mathcal{L}_{\text{KROPE}}(\omega)$$
$$:= \mathbb{E}_\mathcal{D}\left[\left(\mathcal{K}_{\bar{\omega}}(s_1, a_1; s_2; a_2) - \tilde{k}_\omega(s_1, a_1; s_2, a_2)\right)^2\right],$$
$$(4)$$

where $\mathcal{K}_{\bar{\omega}}(s_1, a_1; s_2; a_2) = 1 - \frac{|r(s_1,a_1)-r(s_2,a_2)|}{|r_{\max}-r_{\min}|} + \gamma \mathbb{E}_{\pi_e}[\tilde{k}_{\bar{\omega}}(s_1', a_1'; s_2', a_2')]$, the state-action pairs are sampled from $\mathcal{D}$, and $\bar{\omega}$ are weights of the target network that are periodically copied from $\omega$ (Mnih et al., 2015). In this work, we use KROPE as an auxiliary task, which introduces only an auxiliary task weight as the additional hyperparameter. It is critical to note that since the learning algorithm is a semi-gradient method, it may still diverge. Nevertheless, in Section 4 we show KROPE still improves the stability and accuracy of OPE compared to baselines.

# 4. Empirical Results

In this section, we present our empirical study designed to answer the following question: **do KROPE representations lead to stable MSVE and low MSVE?**

## 4.1. Empirical Setup

In this section, we describe the main details of our empirical setup. For further details such as datasets, policies, hyperparameters, and evaluation protocol please refer to Appendix C.

**Baselines** Our primary representation learning baseline is fitted q-evaluation (FQE) (Le et al., 2019). FQE is the most fundamental deep RL OPE algorithm that learns representations of state-actions to predict the long-term performance of a policy. While FQE is typically used as an OPE algorithm, it can also be viewed as a value-predictive representation learning algorithm (Lehnert & Littman, 2020). More specifically, consider its loss function: $\mathbb{E}_{(s,a,s')\sim\mathcal{D}}\left[\left(r(s,a) + \gamma\mathbb{E}_{a'\sim\pi_e}[q_{\bar{\xi}}(s',a')] - q_{\xi}(s,a)\right)^2\right]$, where $q_{\xi}(s,a) := \phi_{\xi'}(s,a)^{\top}w$ and $\xi = \{\xi', w\}$. We view $\xi$ as the neural network weights of an action-value neural network and $w$ as the linear weights of the network applied on the output of the penultimate layer $\phi_{\xi'}(s,a)$ of the neural network. Then minimizing this loss function shapes the representations $\phi_{\xi'}(s,a)$ to predict the expected future discounted return. As noted in Section 2, we follow the linear evaluation protocol where $\phi_{\xi'}$ is shaped by different auxiliary tasks and is then used with LSPE for OPE since it helps us understand the properties of the representations within the context of a well-understood value function learning algorithm (Grill et al., 2020; Chang et al., 2022; Farebrother et al., 2024; Wang et al., 2021). We provide the pseudocode of this setup in Appendix A. We also note that in Appendix C, we present results of performing OPE using FQE instead LSPE, and find that KROPE still reliably produces stable OPE estimates.

We consider the following three classes (and total 6) auxiliary representation learning algorithms that are typically paired with FQE for stability: I) bisimulation-metric based,

II) model-based, and III) co-adaptation penalty-based. In class I, we consider 1) KROPE (ours), 2) deep bisimulation for control (DBC) (Zhang et al., 2021), 3) representations for OPE (ROPE) (Pavse & Hanna, 2023a; Castro et al., 2021); in class II, we consider bellman complete representation learning (BCRL-EXP-NA) (Chang et al., 2022); and in class III, we consider 1) absolute DR3 regularizer (Kumar et al., 2022; Ma et al., 2024) and 2) BEER regularizer (He et al., 2024). In all cases, the penultimate layer features of FQE's action-value encoder $\phi_{\xi'}$ are fed into LSPE for OPE. Note that since BCRL was not designed as an auxiliary task (Chang et al., 2022), we evaluate it as a non-auxiliary (NA) task algorithm. We provide additional details on the baselines in Appendix C.

**Domains** We conduct our evaluation on a variety of domains: 1) Garnet MDPs, which are a class of tabular stochastic MDPs that are randomly generated given a fixed number of states and actions (Archibald et al., 1995); 2) 4 DM Control environments: CartPoleSwingUp, CheetahRun, FingerEasy, WalkerStand (Tassa et al., 2018); and 3) 9 D4RL datasets (Fu et al., 2020; 2021). The first domain enables us to analyze the algorithms' performance across a wide range of stochastic tabular MDPs. The second and third set of domains test the algorithms in continuous higher-dimensional state-action environments.

## 4.2. Analyzing Stability and $q^{\pi_e}$-Consistency Properties

In this set of experiments on the Garnet MDPs domain, we analyze the stability and $q^{\pi_e}$ consistency properties of the learned representations. We present the results in Figure 2. Our Garnet MDPs were generated with 8 states and 5 actions, with a total of $|\mathcal{X}| = 40$ state-actions, and each native $(s, a)$ representation is a 1-hot vector. In these experiments, the native representation is fed into a linear encoder with a bias component and no activation function. All algorithms are trained for 500 epochs and we report the results by evaluating the final learned representations for different latent dimensions $d$.

**Stability.** Based on Theorem 1, a representation is stable if it induces a spectral radius of $(\mathbb{E}[\Phi^{\top}\Phi])^{-1}(\gamma\mathbb{E}[\Phi^{\top}P^{\pi_e}\Phi])$ that is less than 1. In Figure 2(a), we present the fraction of runs that result in such representations. We find that up till $d = 30$, 100% of KROPE and BEER runs have spectral radius less than 1. We also find that BCRL-EXP-NA produces stable representations up till $d = 40$. At $d = 50$, all algorithms produce unstable representations. These results suggest that KROPE, BEER, and BCRL-EXP-NA are reliable in producing stable representations when projecting state-actions into low dimensions. When $d \geq |\mathcal{X}|$, the covariance matrix $\mathbb{E}[\Phi^{\top}\Phi]$ is more likely to be a singular matrix, which implies higher chance of instability.

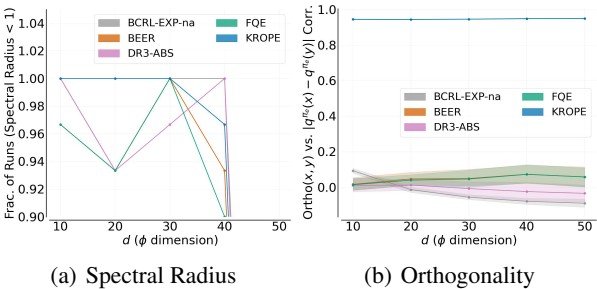

(a) Spectral Radius      (b) Orthogonality

*Figure 2.* Evaluation of basic representation properties on Garnet MDPs with 40 state-actions vs. output dimension $d$. Figure 2(a): Fractions of runs out of 30 trials that resulted in spectral radius of $(\mathbb{E}[\Phi^\top \Phi])^{-1}(\gamma\mathbb{E}[\Phi^\top P^{\pi_e}\Phi])$ to be less than 1; higher is better. Figure 2(b): Pearson correlation between orthogonality between pairs of latent features vs. their corresponding absolute $q^{\pi_e}$ action-value difference; higher is better. All results are averaged over 30 trials and the shaded region is the 95% confidence interval.

$q^{\pi_e}$-**Consistency.** We say that the representations have maintained $q^{\pi_e}$-consistency well when state-actions that have similar $q^{\pi_e}$ values are close to each other in the representation space (Lan et al., 2021). We assess this property by first measuring the orthogonality: $1 - \frac{|\langle\phi(s_1,a_1),\phi(s_2,a_2)\rangle|}{\|\phi(s_1,a_1)\|\|\phi(s_2,a_2)\|}$ (Wang et al., 2024) between every state-action pair, $(s_1, a_1; s_2, a_2)$, and the absolute action-value difference: $|q^{\pi_e}(s_1, a_1) - q^{\pi_e}(s_2, a_2)|$. We then compute the Pearson correlation between these values for every pair and plot the correlation for each $d$ in Figure 2(b). A correlation coefficient close to 1 indicates that the representations maintain $q^{\pi_e}$-consistency well. We find that KROPE representations satisfy this property almost perfectly since it specifically tries to learn representations such that state-action pairs with similar values under $\pi_e$ are similar. We observe that the other non-bisimulation-based algorithms typically have zero or even negative correlation. A negative correlation indicates that state-actions with different action-values may have similar representations in latent space, which can complicate value function learning.

### 4.3. Offline Policy Evaluation

In this set of experiments, we evaluate the algorithms for OPE on 13 datasets: 4 DM control and 9 D4RL datasets. We also evaluate BCRL-NA, which is BCRL without the exploration maximization regularizer. To stabilize training for all algorithms, we use wide neural networks with layernorm (Gallici et al., 2025; Ota et al., 2024). Note that while wide networks and layernorm stabilize training, they may not lead to stable LSPE under the linear evaluation protocol. We report the final (normalized) squared value errors when the learned representations are used with LSPE for OPE. The results are presented in Table 1 (see Appendix C.3.1 for full

learning curves).

In general, we find that KROPE representations lead to low and stable MSVE on $10/13$ datasets. On the other hand, we find that the other auxiliary tasks inconsistently produce stable OPE estimates across all environments. Compared to the other bisimulation-based algorithms (DBC and ROPE), KROPE consistently achieves lower error. In all cases, DBC seems to be unreliable for OPE. ROPE, which was the previous state-of-the-art bisimulation-based algorithm for OPE, also achieves low error. While ROPE is competitive with KROPE, KROPE generally outperforms ROPE and has one less hyperparameter. The performance of the ABS-DR3 and BEER regularizer suggests that explicitly trying to increase the rank of the features of the penultimate layer may hurt stability, and even if the OPE error is stable, it can hurt accuracy. We also make a similar observation for BCRL. However, in this case, we attribute poor performance to difficulty in optimizing the BCRL objective. In fact, in Figure 3(a), we will see that BCRL is sensitive to hyperparameter tuning. We also observe results consistent with a known result that BCRL-EXP-NA achieves lower error than BCRL-NA indicating the known result that exploration maximization of the covariance matrix helps produce stable representations (Chang et al., 2022). While FQE achieves low error on WalkerStand and CheetahRun, it is very unstable on the other datasets, which motivates the need to shape the representations for stable and accurate OPE. Note that in practice, KROPE may still diverge, as it did it in 3 cases, because the learning objective in Equation (4) is still a semi-gradient learning algorithm (see discussion in Section 5 and Section 4.4).

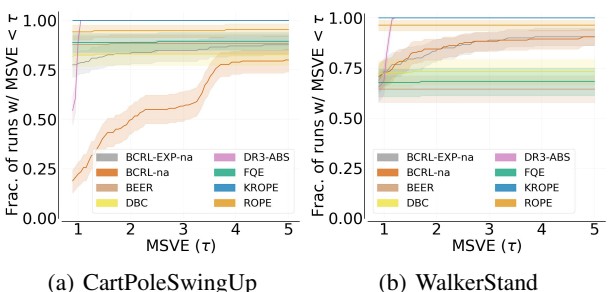

(a) CartPoleSwingUp      (b) WalkerStand

*Figure 3.* Hyperparameter sensitivity plots on CartPoleSwingUp and WalkerStand. Results are over 20 trials for each hyperparameter configuration and shaded region is the 95% confidence interval. Larger area under the curve is better.

**Hyperparameter Sensitivity.** In OPE, hyperparameter tuning can be challenging since it may be infeasible to get access to ground truth performance of $\pi_e$ (Fu et al., 2021). Therefore, we prefer algorithms that are robust to hyperparameter tuning, i.e, they reliably produce accurate OPE estimates for a wide range of hyperparameters. In Figure 3, we present the performance profile for each algorithm

| | Algorithm | | | | | | | |
|---|---|---|---|---|---|---|---|---|
| Dataset (DMC) | FQE | BCRL+EXP | BCRL | BEER | DR3 | DBC | ROPE | KROPE (ours) |
| CartPoleSwingUp | Div. | $2.0 \pm 1.6$ | $2.2 \pm 0.8$ | Div. | $0.9 \pm 0.0$ | Div. | $0.2 \pm 0.1$ | $\mathbf{0.0 \pm 0.0}$ |
| CheetahRun | $\mathbf{0.0 \pm 0.0}$ | $0.3 \pm 0.2$ | $0.8 \pm 0.3$ | $\mathbf{0.0 \pm 0.0}$ | $0.4 \pm 0.0$ | Div. | Div. | $\mathbf{0.0 \pm 0.0}$ |
| FingerEasy | Div. | $0.6 \pm 0.1$ | $0.8 \pm 0.2$ | Div. | $0.9 \pm 0.0$ | Div. | $\mathbf{0.1 \pm 0.0}$ | $0.6 \pm 0.0$ |
| WalkerStand | $\mathbf{0.0 \pm 0.0}$ | $0.2 \pm 0.2$ | $0.2 \pm 0.1$ | $1.9 \pm 3.6$ | $0.1 \pm 0.0$ | Div. | $0.2 \pm 0.0$ | $\mathbf{0.0 \pm 0.0}$ |
| Dataset (D4RL) | FQE | BCRL+EXP | BCRL | BEER | DR3 | DBC | ROPE | KROPE (ours) |
| cheetah random | $\mathbf{0.9 \pm 0.0}$ | Div. | Div. | $\mathbf{0.9 \pm 0.0}$ | $\mathbf{0.9 \pm 0.0}$ | $\mathbf{0.9 \pm 0.0}$ | $1.0 \pm 0.0$ | $1.0 \pm 0.0$ |
| cheetah medium | Div. | Div. | $0.2 \pm 0.2$ | Div. | Div. | Div. | $\mathbf{0.0 \pm 0.0}$ | $\mathbf{0.0 \pm 0.0}$ |
| cheetah med-expert | Div. | $0.2 \pm 0.1$ | $0.3 \pm 0.1$ | Div. | Div. | Div. | $0.1 \pm 0.0$ | $\mathbf{0.0 \pm 0.0}$ |
| hopper random | Div. | Div. | Div. | Div. | $0.8 \pm 0.0$ | Div. | Div. | $\mathbf{0.1 \pm 0.0}$ |
| hopper medium | Div. | Div. | Div. | Div. | Div. | Div. | Div. | Div. |
| hopper med-expert | Div. | Div. | Div. | Div. | $0.6 \pm 0.0$ | Div. | $\mathbf{0.0 \pm 0.0}$ | $\mathbf{0.0 \pm 0.0}$ |
| walker random | Div. | Div. | Div. | Div. | $1.0 \pm 0.0$ | Div. | Div. | $\mathbf{0.5 \pm 0.1}$ |
| walker medium | Div. | Div. | Div. | Div. | Div. | Div. | Div. | Div. |
| walker med-expert | Div. | $1.3 \pm 0.4$ | $2.6 \pm 2.1$ | Div. | $6.6 \pm 11.6$ | Div. | $\mathbf{0.1 \pm 0.0}$ | Div. |

*Table 1.* Final normalized squared value error achieved by LSPE with different representation learning algorithms. Results are averaged over 20 trials and the variation is the 95% confidence interval. Lower and less erratic is better. Values are rounded to single place decimal. When an algorithm's error is $\geq 10$, we label it as diverged (Div.). **Bolded and highlighted** error indicates lowest error among baselines.

across *all* hyperparameter combinations and *all* trials (Agarwal et al., 2021). We tune the hyperparameters discussed in Appendix C.1. We find that $100\%$ KROPE runs across all instances produce MSVE $\leq 1$, which is not the case with other algorithms. This result indicates the reliability of using KROPE for OPE over other algorithms including other bisimulation-based algorithms.

> **Takeaway #2: Practical Offline Policy Evaluation with KROPE**
>
> KROPE can *increase* the stability and accuracy of evaluation of offline RL agents.

### 4.4. Stability of the KROPE Learning Procedure

In this section, we compare the susceptibility of KROPE's and FQE's semi-gradient learning to divergence. We refer the reader to Appendix C.1 for more details.

We conduct our experiments on the Markov reward process (MRP) in Figure 4(a) (see Feng et al. (2019)). The MRP consists of 4 non-terminal states, 1 terminal state (the box), and only 1 action. The value function estimate is linear in the weights $w = [w_1, w_2, w_3]$, so, starting clockwise from the left-most circle, the native features of the states are $[1, 0, 0]$, $[0, 1, 0]$, $[0, 0, 2]$, and $[0, 0, 1]$. In this setup, we say a transition is *bad* if the bootstrapping target is a moving target for the current state. For example, the transition from

$w_3$ to $2w_3$ is a bad transition since updates made to $w_3$ may move $2w_3$ further away. When this transition is sampled at a frequency that is different from the on-policy distribution, algorithms such as TD, LSPE, and FQE tend to diverge (Asadi et al., 2024). Similarly, for KROPE, which uses its weights to compute the latent representation of states, we would expect that *pairs* of transitions that lead to moving dot product targets are bad transition pairs.

To understand the stability of the learning procedures, we design the following experiment when using KROPE as an auxiliary loss function to FQE. We first create an on-policy dataset $\mathcal{D}^{\mathrm{on}}$. We then define off-policy datasets of the form $\mathcal{D}^s$, which consists of transitions starting from the specified state $s$, where $s = \{w_1, w_2, w_3, 2w_3\}$. Using these datasets, we then construct $\mathcal{D}_1 = \mathcal{D}^{\mathrm{on}} \cup \mathcal{D}^{w_3}$ and four $\mathcal{D}_2$ variations: 1) $\mathcal{D}_2^{w_1} = \mathcal{D}^{\mathrm{on}} \cup \mathcal{D}^{w_1}$, 2) $\mathcal{D}_2^{w_2} = \mathcal{D}^{\mathrm{on}} \cup \mathcal{D}^{w_2}$, 3) $\mathcal{D}_2^{w_3} = \mathcal{D}^{\mathrm{on}} \cup \mathcal{D}^{w_3}$, and 4) $\mathcal{D}_2^{2w_3} = \mathcal{D}^{\mathrm{on}} \cup \mathcal{D}^{2w_3}$. Therefore, $\mathcal{D}_1, \mathcal{D}_2^{w_3}, \mathcal{D}_2^{2w_3}$ consist of mostly bad transitions while $\mathcal{D}_2^{w_1}, \mathcal{D}_2^{w_2}$ consist of good transitions. Since KROPE samples *pairs* of transitions, we investigate the consequences of pairing $\mathcal{D}_1$ with a $\mathcal{D}_2$ variation. We pair each of the four $\mathcal{D}_2$ datasets with $\mathcal{D}_1$ and train FQE+KROPE, where FQE is fed transition samples from $\mathcal{D}_1$ and KROPE is fed samples from $\mathcal{D}_1$ and the specific $\mathcal{D}_2$ variation.

In Figure 4(b), we show the training loss of only KROPE to analyze when KROPE may diverge. We find that even though $\mathcal{D}_1$ consists of mostly bad transitions, if KROPE sam-

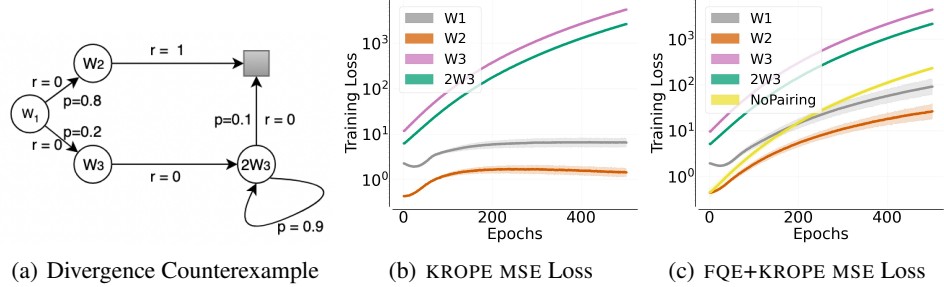

(a) Divergence Counterexample      (b) KROPE MSE Loss      (c) FQE+KROPE MSE Loss

*Figure 4.* Figure 4(a): MRP counterexample designed to illustrate divergence; $r$ denotes the rewards and $p$ denotes the probability of transition (Feng et al., 2019). Figures 4(b) and 4(c): KROPE training loss and FQE+KROPE training loss vs. epochs respectively when different datasets are paired with $\mathcal{D}_1$; results are averaged over 20 trials, shaded region is the 95% confidence interval, and lower is better.

ples good transitions from $\mathcal{D}_2^{w_1}$ or $\mathcal{D}_2^{w_2}$, its divergence is mitigated. However, if KROPE samples bad transitions from $\mathcal{D}_2^{w_3}$ or $\mathcal{D}_2^{2w_3}$, it diverges since the pairing of samples from $(\mathcal{D}_1, \mathcal{D}_2^{w_3})$ and $(\mathcal{D}_1, \mathcal{D}_2^{2w_3})$ leads to KROPE chasing a moving bootstrapped dot product target. Therefore, sampling *pairs* of bad transitions can make KROPE more likely to diverge, while sampling a single bad transition (with a good transition) can make it less likely to diverge.

In Figure 4(c), we plot the combined training loss of FQE+KROPE. As expected, when FQE uses only bad transitions from $\mathcal{D}_1$, it diverges (No Pairing). In fact, in all cases, the divergence is due to FQE even when the corresponding KROPE variation is not diverging (see $\mathcal{D}_2^{w_1}$ or $\mathcal{D}_2^{w_2}$ in Figure 4(b)). For the $\mathcal{D}_2^{w_3}$ or $\mathcal{D}_2^{2w_3}$ variations, both FQE and KROPE diverge. Therefore, sampling single bad transitions makes FQE more likely to diverge (Asadi et al., 2024).

While we hand-designed these datasets, in general, we would expect that the probability of sampling a *pair* of bad transitions is less than that of sampling a single bad transition[1]. These experiments show that while value function learning with FQE may diverge, representation learning with KROPE may not.

> **Takeaway #3: KROPE Divergence**
>
> While FQE's divergence is due to sampling bad transitions, KROPE's divergence is due to sampling bad *pairs* of transitions. Intuitively, since the probability of sampling a bad transition *pair* is less than that of sampling a single bad transition, KROPE training may be easier than FQE training.

## 5. Limitations and Future Work

In this section, we discuss limitations and future work. A shortcoming of KROPE's semi-gradient algorithm is its sus-

ceptibility to divergence (Sutton & Barto, 2018). While we employed commonly-used techniques such as layer-norm and wide neural networks to mitigate divergence (Ota et al., 2024; Gallici et al., 2025), the consequences of a semi-gradient method may still exist. One potential remedy is to avoid semi-gradient learning. Feng et al. (2019) suggest to replace the fixed-point loss function of semi-gradient methods with an equivalent expression that leverages the Legendre-Fenchel transformation (Rockafellar & Wets, 1998). However, a drawback with this approach is that the new learning objective is a minimax procedure, which can be challenging to optimize in practice. In future work, we will explore the viability of this approach to design a provably convergent version of KROPE.

## 6. Conclusion

In this work, we tackled the problem of stabilizing offline value function learning in reinforcement learning. We introduced a bisimulation-based representation learning algorithm, kernel representations for OPE (KROPE), that shapes the state-action representations to stabilize this procedure. Theoretically, we showed that KROPE representations are stable from two perspectives: 1) non-expansiveness, i.e., they lead to value function learning updates that enable convergence to a fixed-point and 2) Bellman completeness, i.e., they satisfy a condition for data-efficient policy evaluation. Empirically, we showed that KROPE leads to more stable and accurate offline value function learning than baselines. We also demonstrated when representation learning with KROPE may be easier than value function learning with FQE. Our work showed that bisimulation-based representation learning can improve the stability and accuracy of long-term performance evaluations of offline reinforcement learning agents.

---

[1]We expect this statement to roughly hold true unless the data consists of mostly bad transitions; see Appendix C.1

## Impact Statement

Our work is largely focused on studying fundamental RL research questions, and thus we do not see any immediate negative societal impacts.

## Acknowledgments

The authors thank Abhinav Narayan Harish, Adam Labiosa, Andrew Wang, Anshuman Senapati, David Abel, Kyle Domico, Ishan Durugkar, Pablo Samuel Castro, Prabhat Nagarajan, Tengyang Xie, and the anonymous reviewers at ICLR and ICML for feedback on our work, and thank Fanghui Liu for inspiring discussions on kernel methods. Y. Chen is partially supported by NSF CCF-2233152. Q. Xie is supported in part by National Science Foundation (NSF) grants CNS-1955997, EPCN-2339794 and EPCN-2432546. J. Hanna acknowledges support from NSF (IIS-2410981), American Family Insurance through a research partnership with the University of Wisconsin—Madison's Data Science Institute, and Sandia National Labs through a University Partnership Award.

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

# A. Background

In this section, we present the theoretical background.

## A.1. Bisimulation Metrics

In this section, we present background information on bisimulations and its associated metrics. Our proposed representation learning algorithm is a bisimulation-based algorithm. Bisimulation abstractions are those under which two states with identical reward functions and that lead to identical groups of next states under any action are classified as similar (Ferns et al., 2004; 2011; Ferns & Precup, 2014). Bisimulations are the strictest forms of abstractions. In practice, the exact bisimulation criterion is difficult to satisfy computationally and statistically. A more relaxed version of this notion is the *π-bisimulation metrics*. These metrics capture the similarity between two states such that two states are considered similar if they have identical *expected* reward functions and *expected* transitions to identical groups of next states under a *fixed policy* π (Castro, 2020).

We first give the definition of bisimulation.

**Definition 3.** *(Li et al., 2006) An abstraction $\phi : \mathcal{S} \to \mathcal{S}^\phi$ over the state space $\mathcal{S}$ is a bisimulation if for any action $a$ and any abstract state $s^\phi \in \mathcal{S}^\phi$, $\phi$ is such that for any two states $s_1, s_2 \in \mathcal{X}$, $\phi(s_1) = \phi(s_2)$ implies that $r(s_1, a) = r(s_2, a)$ and $\sum_{s' \in s^\phi} P(s'|s_1, a) = \sum_{s' \in s^\phi} P(s'|s_2, a)$.*

Below we define π-bisimulations for state-actions instead of states:

**Definition 4.** *(Castro, 2020) An abstraction $\phi : \mathcal{X} \to \mathcal{X}^\phi$ over the state-action space $\mathcal{X}$ is a π-bisimulation for a fixed policy π if for any two state-actions $x, y \in \mathcal{X}$ and abstract state-action $x^\phi \in \mathcal{X}^\phi$, $\phi$ is such that $\phi(x) = \phi(y)$ implies that $r(x) = r(y)$ and $\sum_{x' \in x^\phi} P^\pi(x'|x) = \sum_{x' \in x^\phi} P^\pi(x'|y)$.*

The above definitions are based on exact groupings between state-actions. This strictness motivates the use of bisimulation and π-bisimulation metrics, which we define below.

**Theorem 3.** *(Ferns et al., 2004) Let $\mathcal{M}(\mathcal{S})$ be the space of bounded pseudometrics on the state-space $\mathcal{S}$. Then define $\mathcal{B} : \mathcal{M}(\mathcal{S}) \to \mathcal{M}(\mathcal{S})$ such that for each $d \in \mathcal{M}(\mathcal{S})$:*

$$\mathcal{B}(d)(s_1, s_2) = \max_{a \in \mathcal{A}}(|r(s_1, a) - r(s_2, a)| + \gamma \mathcal{W}(d)(P(\cdot|s_1, a), P(\cdot|s_2, a)),$$

*where $\mathcal{W}$ is Wasserstein distance between the two distributions under metric $d$. Then $\mathcal{B}$ has a unique fixed point, $d^*$, and $d^*$ is a bisimulation metric.*

Similarly, we have the π-bisimulation metric:

**Theorem 4.** *(Castro, 2020) Let $\mathcal{M}(\mathcal{X})$ be the space of bounded pseudometrics on the state-action space $\mathcal{X}$ and π be a fixed policy. Then define $\mathcal{B} : \mathcal{M}(\mathcal{X}) \to \mathcal{M}(\mathcal{X})$ such that for each $d \in \mathcal{M}(\mathcal{S})$:*

$$\mathcal{B}(d)(x, y) = |r(x) - r(y)| + \gamma \mathcal{W}(d)(P^\pi(\cdot|x), P^\pi(\cdot|y)),$$

*where $\mathcal{W}$ is Wasserstein distance between the two distributions under metric $d$. Then $\mathcal{B}$ has a unique fixed point, $d^*$, and $d^*$ is a π-bisimulation metric.*

Using the above metrics, prior works have introduced several representation learning algorithms to learn representations such that the distance between representations in latent space model the above distance metrics (Castro et al., 2021; 2023; Zhang et al., 2021; Kemertas & Aumentado-Armstrong, 2021; Pavse & Hanna, 2023a).

## A.2. Reproducing Kernel Hilbert Spaces

Let $\mathcal{X}$ be a finite set and define a function $k : \mathcal{X} \times \mathcal{X} \to \mathbb{R}$ to be a positive semidefinite kernel if it is symmetric and positive semidefinite. We then have for any $\{x_1, x_2, ..., x_n\} \in \mathcal{X}$ and $\{c_1, c_2, ..., c_n\} \in \mathbb{R}$:

$$\sum_{i=1}^n \sum_{j=1}^n c_i, c_j k(x_i, x_j) \geq 0$$

Note that if the above inequality is strictly greater than zero whenever $\{c_1, \ldots, c_n\}$ has at least one nonzero, we say the kernel is positive definite. Given a kernel $k$ on $\mathcal{X}$ with the reproducing property, we can construct a space of functions $\mathcal{H}_k$ referred to as a reproducing kernel Hilbert space (RKHS) with the following steps:

1. Construct a vector space of real-valued functions on $\mathcal{X}$ of the form $\{k(x, \cdot) : x \in \mathcal{X}\}$.

2. Equip this space with an inner product given by $\langle k(x, \cdot), k(y, \cdot) \rangle_{\mathcal{H}_k} = k(x, y)$.

3. Take the completion of the vector space with respect to the above inner product.

Our resulting vector space $\mathcal{H}_k$ is then an RKHS.

It is often convenient to write $\psi(x) := k(x, \cdot) \in \mathcal{H}_k$, which is called the feature map and is an embedding of $x$ in $\mathcal{H}_k$. One can also embed probability distributions into $\mathcal{H}_k$. That is, $\Phi : \mathcal{P}(\mathcal{X}) \to \mathcal{H}_k$, which maps probability distributions over $\mathcal{X}$ to $\mathcal{H}_k$. We define $\Phi(\mu) = \mathbb{E}_{X \sim \mu}[\psi(X)]$, which is the mean embedding in $\mathcal{H}_k$ under $\mu$.

Given these embeddings in the Hilbert space, we can quantify the distances between elements in $\mathcal{X}$ and $\mathcal{P}(\mathcal{X})$ in terms of the embeddings.

**Definition 5.** *Given a positive semidefinite kernel $k$, define $\rho_k$ as its induced distance:*

$$\rho_k := \|\psi(x) - \psi(y)\|_{\mathcal{H}_k}.$$

*By expanding the inner product, the squared distance can be written in terms of $k$:*

$$\rho_k^2(x, y) = k(x, x) + k(y, y) - 2k(x, y).$$

Similarly, we have distances on $\mathcal{P}(\mathcal{X})$ using $\Phi$:

**Definition 6.** *(Gretton et al., 2012) Let $k$ be a kernel on $\mathcal{X}$ and $\Phi : \mathcal{P}(\mathcal{X}) \to \mathcal{H}_k$ be as defined above. Then the Maximum Mean Discrepancy (MMD) is a pseudo metric on $\mathcal{P}(\mathcal{X})$ defined by:*

$$MMD(k)(\mu, \nu) = \|\Phi(\mu) - \Phi(\nu)\|_{\mathcal{H}_k}.$$

The core usage of the RKHS is to precisely characterize the nature of the KROPE kernel. In practice, we deal with neural network representations, which are embedded in Euclidean space. Therefore, our goal is to learn representations in Euclidean space that approximate the properties of representations in the RKHS. For more details on the RKHS, we refer readers to (Castro et al., 2023) and (Gretton et al., 2012).

### A.3. Algorithm Pseudocode

In this section, we present the pseudocode for LSPE and for our FQE + auxiliary task with LSPE for OPE setup.

---

**Algorithm 1** LSPE

---

1: Input: policy to evaluate $\pi_e$, batch $\mathcal{D}$, fixed encoder function $\phi : \mathcal{S} \times \mathcal{A} \to \mathbb{R}^d$.
2: Initialize $\theta_0 \in \mathbb{R}^d$ randomly.
3: Apply $\phi$ to $\mathcal{D}$ to generate $\Phi$.
4: **for** t = 0, 1, 2, ... $T - 1$ **do**
5:     $\theta_{t+1} \leftarrow (\mathbb{E}[\Phi^\top \Phi])^{-1} \mathbb{E}[\Phi^\top (r + \gamma P^{\pi_e} \Phi \theta_t)]$
6: **end for**
7: Return $\theta_T$

---

---

**Algorithm 2** FQE + representation learning auxiliary task with LSPE for OPE

---

1: Input: policy to evaluate $\pi_e$, batch $\mathcal{D}$, encoder parameters class $\Omega$, encoder function $\phi : \mathcal{S} \times \mathcal{A} \to \mathbb{R}^d$, action-value linear function $q : \mathbb{R}^d \to \mathbb{R}$, $\alpha \in [0, 1]$.

2: **for** epoch = 1, 2, 3, ... T **do**

3:    $\mathcal{L}(\omega) := \alpha\text{Aux-Task}(\phi_\omega, \mathcal{D}, \pi_e) + (1 - \alpha)\mathbb{E}_{(s,a,s')\sim\mathcal{D}}\left[\left(r(s, a) + \gamma\mathbb{E}_{a'\sim\pi_e}[q_{\bar{\xi}}(\phi_{\hat{\omega}}(s', a'))] - q_\xi(\phi_{\hat{\omega}}(s, a))\right)^2\right]$
   {where the penultimate features $\phi$ are fed into an auxiliary representation learning algorithm such as KROPE, DR3, BEER etc.}

4:    $\hat{\omega}_t := \arg\min_{\omega\in\Omega}\mathcal{L}(\omega)$

5:    Periodically run LSPE, $\theta := \text{LSPE}(\pi_e, \mathcal{D}, \phi_\omega)$.

6:    Compute estimated action-values, $\hat{q} := \Phi_{\hat{\omega}_t}\theta$. {where $\phi_{\hat{\omega}}$ is applied to $\mathcal{D}$ to get $\Phi_\omega$}

7: **end for**

8: Return $\hat{q} := \Phi_{\hat{\omega}_T}\theta$. {Estimated action-value function of $\pi_e$, $q^{\pi_e}$.}

---

## B. Theoretical Results

In this section, we present the proofs of our main and supporting theoretical results. The first set of proofs in Section B.1 show that KROPE is a valid operator. While new to our work, the proofs follow those by (Castro et al., 2023). The next set of proofs in Section B.2 prove the stability of KROPE representations and are novel to our work. For presentation purposes, it will often be convenient to refer to a state-action pair as $x \in \mathcal{X}$ instead of $(s, a)$.

### B.1. KROPE Operator Validity

Given the definition of the KROPE kernel, we now present an operator that converges to $k^{\pi_e}$:

**Definition 7** (KROPE operator). *Given a target policy $\pi_e$, the KROPE operator $\mathcal{F}^{\pi_e} : \mathbb{R}^{\mathcal{X}\times\mathcal{X}} \to \mathbb{R}^{\mathcal{X}\times\mathcal{X}}$ is defined as follows: for each kernel $k : \mathcal{X} \times \mathcal{X} \to \mathbb{R}$, $\forall(s_1, a_1; s_2, a_2) \in \mathcal{X} \times \mathcal{X}$,*

$$\mathcal{F}^{\pi_e}(k)(s_1, a_1; s_2, a_2) := \underbrace{k_1(s_1, a_1; s_2, a_2)}_{\text{short-term similarity}} + \gamma\underbrace{\mathbb{E}_{s'_1, s'_2\sim P, a'_1, a'_2\sim\pi_e}[k(s'_1, a'_1; s'_2, a'_2)]}_{\text{long-term similarity}} \tag{5}$$

*where $s'_1 \sim P(s'_1|s_1, a_1), s'_2 \sim P(s'_2|s_2, a_2), a'_1 \sim \pi_e(\cdot|s'_1), a'_2 \sim \pi_e(\cdot|s'_2)$, and $k_1(s_1, a_1; s_2, a_2) := 1 - \frac{|r(s_1, a_1) - r(s_2, a_2)|}{|r_{max} - r_{min}|}$ is a positive semidefinite kernel.*

We now present the proofs demonstrating the validity of the KROPE operator. All the proofs in this sub-section model those by (Castro et al., 2023). The primary difference is that our operator is for state-actions instead of states.

**Lemma 1.** *Let $\mathcal{K}(\mathcal{X})$ be the space of positive semidefinite kernels on $\mathcal{X}$. The KROPE operator $\mathcal{F}^{\pi_e}$ is a contraction with modulus $\gamma$ in $\|\cdot\|_\infty$.*

*Proof.* Let $k_1, k_2 \in \mathcal{K}(\mathcal{X})$. We then have:

$$\|\mathcal{F}^{\pi_e}(k_1) - \mathcal{F}^{\pi_e}(k_2)\|_\infty$$
$$= \max_{(x,y)\in\mathcal{X}\times\mathcal{X}}|\mathcal{F}^{\pi_e}(k_1)(x, y) - \mathcal{F}^{\pi_e}(k_2)(x, y)|$$
$$= \gamma\max_{(x,y)\in\mathcal{X}\times\mathcal{X}}|\mathbb{E}_{X'\sim P^{\pi_e}(\cdot|x),Y'\sim P^{\pi_e}(\cdot|y)}[k_1(X', Y')] - \mathbb{E}_{X'\sim P^{\pi_e}(\cdot|x),Y'\sim P^{\pi_e}(\cdot|y)}[k_2(X', Y')]|$$
$$= \gamma\max_{(x,y)\in\mathcal{X}\times\mathcal{X}}|\mathbb{E}_{X'\sim P^{\pi_e}(\cdot|x),Y'\sim P^{\pi_e}(\cdot|y)}[k_1(X', Y') - k_2(X', Y')]|$$
$$\leq \gamma\|k_1 - k_2\|_\infty.$$

This completes the proof of the lemma. $\square$

**Lemma 2.** *Let $\mathcal{K}(\mathcal{X})$ be the space of positive semidefinite kernels on $\mathcal{X}$. Then the metric space $(\mathcal{K}(\mathcal{X}), \|\cdot\|_\infty)$ is complete.*

*Proof.* To show that $\mathcal{K}(\mathcal{X})$ is complete it suffices to show that every Cauchy sequence $\{k_n\}_{n\geq0}$ has a limiting point in $\mathcal{K}(\mathcal{X})$. Since $\mathcal{X}$ is a finite, the space of function $\mathbb{R}^{\mathcal{X}\times\mathcal{X}}$ is a finite-dimensional vector space, which is complete under $\|\cdot\|_\infty$.

Thus, the limiting point $k \in \mathbb{R}^{\mathcal{X} \times \mathcal{X}}$ of the Cauchy sequence $\{k_n\}_{n \geq 0}$ lies in $\mathbb{R}^{\mathcal{X} \times \mathcal{X}}$. Moreover, since we are considering only positive semidefinite kernel elements in the Cauchy sequence and they uniformly converge to $k \in \mathbb{R}^{\mathcal{X} \times \mathcal{X}}$, $k$ must also be positive semidefinite. Thus, $\mathcal{K}(\mathcal{X})$ is complete under $\| \cdot \|_\infty$. $\qquad \square$

**Proposition 2.** *The* KROPE *operator* $\mathcal{F}^{\pi_e}$ *has a unique fixed point in* $\mathcal{K}(\mathcal{X})$. *That is, there is a unique kernel* $k^{\pi_e} \in \mathcal{K}(\mathcal{X})$ *satisfying*

$$k^{\pi_e}(s_1, a_1; s_2, a_2) = 1 - \frac{|r(s_1, a_1) - r(s_2, a_2)|}{|r_{max} - r_{min}|} + \gamma \mathbb{E}_{s_1', s_2' \sim P, a_1', a_2' \sim \pi_e}[k^{\pi_e}(s_1', a_1'; s_2', a_2')].$$

*Proof.* Due to Lemmas 1 and 2, $\mathcal{F}^{\pi_e}$ is a contraction in a complete metric space. Therefore, by Banach's fixed point theorem, the unique fixed point $k^{\pi_e}$ exists. $\qquad \square$

**Proposition 3.** *The* KROPE *similarity metric* $d_{\text{KROPE}}$ *satisfies:*

$$\forall x, y \in \mathcal{X}, d_{\text{KROPE}}(x, y) = |r(x) - r(y)| + \gamma \textit{MMD}^2(k^{\pi_e})(P^{\pi_e}(\cdot|x), P^{\pi_e}(\cdot|y)).$$

*Proof.* To see this fact, we can write out the squared Hilbert space distance:

$$\begin{aligned}
d_{\text{KROPE}}(x, y) &= \|\psi^{\pi_e}(x) - \psi^{\pi_e}(y)\|^2_{\mathcal{H}_k^{\pi_e}} \\
&= k^{\pi_e}(x, x) + k^{\pi_e}(y, y) - 2k^{\pi_e}(x, y) \\
&= |r(x) - r(y)| + \gamma \langle \Phi(P^{\pi_e}(\cdot|x)), \Phi(P^{\pi_e}(\cdot|x)) \rangle_{\mathcal{H}_k^{\pi_e}} + \gamma \langle \Phi(P^{\pi_e}(\cdot|y)), \Phi(P^{\pi_e}(\cdot|y)) \rangle_{\mathcal{H}_k^{\pi_e}} \\
&\qquad\qquad\qquad\qquad\qquad\qquad\qquad\qquad - 2\gamma \langle \Phi(P^{\pi_e}(\cdot|x)), \Phi(P^{\pi_e}(\cdot|y)) \rangle_{\mathcal{H}_k^{\pi_e}} \\
&= |r(x) - r(y)| + \gamma \text{MMD}^2(k^{\pi_e})(P^{\pi_e}(\cdot|x), P^{\pi_e}(\cdot|y)),
\end{aligned}$$

where the third line uses

$$k^{\pi_e}(x, x) = \gamma \mathbb{E}_{X_1', X_2' \sim P^{\pi_e}(\cdot|x)}[k^{\pi_e}(X_1', X_2')] = \gamma \langle \Phi(P^{\pi_e}(\cdot|x)), P^{\pi_e}(\cdot|x) \rangle_{\mathcal{H}_k^{\pi_e}}.$$

This completes the proof. $\qquad \square$

Before presenting Lemma 3, we define the distance metric $d_{\text{KROPE}} : \mathcal{X} \times \mathcal{X} \to \mathbb{R}$ induced by the KROPE kernel $k^{\pi_e}$ as follows:

$$\forall x, y \in \mathcal{X} : \ d_{\text{KROPE}}(x, y) := k^{\pi_e}(x, x) + k^{\pi_e}(y, y) - 2k^{\pi_e}(x, y).$$

**Lemma 3.** *We have* $|q^{\pi_e}(x) - q^{\pi_e}(y)| \leq d_{\text{KROPE}}(x, y) + C$, *where* $C = \frac{1}{2} \sum_{n \geq 0} \gamma^n (\Delta_n^{\pi_e}(x) + \Delta_n^{\pi_e}(y))$ *and* $\Delta_n^{\pi_e}(x) = \mathbb{E}_{X' \sim (P^{\pi_e}(\cdot|x))^n}[\mathbb{E}_{X_1'', X_2'' \sim P^{\pi_e}(\cdot|X')}[|r(X_1'') - r(X_2'')|]]$.

*Proof.* We will prove this with induction. We first define the relevant terms involved. We consider the sequences of functions $\{k_m\}_{m \geq 0}$ and $\{q_m\}_{m \geq 0}$, where $k_0, q_0 = 0$. Since $\mathcal{F}^{\pi_e}$ and $\mathcal{T}^{\pi_e}$ are contraction mappings, we know that $\lim_{m \to \infty} k_m = k^{\pi_e}$ and $\lim_{m \to \infty} q_m = q^{\pi_e}$ as $\mathcal{F}^{\pi_e}$ and $\mathcal{T}^{\pi_e}$ are applied respectively at each iteration $m$. At the $m$th application of the operators, we have the corresponding kernel function $k_m$ along with its induced distance function $d_m(x, y) = k_m(x, x) + k_m(y, y) - 2k_m(x, y)$. We will now prove the following for all $m$:

$$|q_m(x) - q_m(y)| \leq d_m(x, y) + \frac{1}{2} \sum_{n \geq 0}^{m} \gamma^n (\Delta_n^{\pi_e}(x) + \Delta_n^{\pi_e}(y)) \tag{6}$$

where $\Delta_n^{\pi_e}(x) = \mathbb{E}_{X' \sim (P^{\pi_e}(\cdot|x))^n}[\mathbb{E}_{X_1'', X_2'' \sim P^{\pi_e}(\cdot|X')}[|r(X_1'') - r(X_2'')|]]$.

The base case $m = 0$ follows immediately since the LHS is zero while the RHS can be non-zero. We now assume the induction hypothesis in Equation (6) is true. We then consider iteration $m + 1$:

$$
\begin{aligned}
&|q_{m+1}(x) - q_{m+1}(y)| \\
&= \left| r(x) + \gamma \mathbb{E}_{X' \sim P^{\pi_e}(\cdot|x)}[q_m(X')] - r(y) - \gamma \mathbb{E}_{Y' \sim P^{\pi_e}(\cdot|y)}[q_m(Y')] \right| \\
&\leq |r(x) - r(y)| + \gamma \mathbb{E}_{X' \sim P^{\pi_e}(\cdot|x), Y' \sim P^{\pi_e}(\cdot|y)}[|q_m(X') - q_m(Y')|] \\
&\leq |r(x) - r(y)| + \gamma \mathbb{E}_{X' \sim P^{\pi_e}(\cdot|x), Y' \sim P^{\pi_e}(\cdot|y)} \left[ d_m(X', Y') + \frac{1}{2} \sum_{n=0}^{m} \gamma^n (\Delta_n^{\pi_e}(X') + \Delta_n^{\pi_e}(Y')) \right] \\
&= |r(x) - r(y)| + \gamma \mathbb{E}_{X' \sim P^{\pi_e}(\cdot|x), Y' \sim P^{\pi_e}(\cdot|y)} \left[ d_m(X', Y') + \frac{1}{2} \sum_{n=1}^{m+1} \gamma^n (\Delta_n^{\pi_e}(x) + \Delta_n^{\pi_e}(y)) \right]
\end{aligned}
$$

where we have used the fact that $\mathbb{E}_{X' \sim P^{\pi_e}(\cdot|x)}[\Delta_n^{\pi_e}(X')] = \Delta_{n+1}^{\pi_e}(x)$. We can then proceed from above as follows:

$$
\begin{aligned}
&= |r(x) - r(y)| + \gamma \mathbb{E}_{X' \sim P^{\pi_e}(\cdot|x), Y' \sim P^{\pi_e}(\cdot|y)} \left[ d_m(X', Y') + \frac{1}{2} \sum_{n=1}^{m+1} \gamma^n (\Delta_n^{\pi_e}(x) + \Delta_n^{\pi_e}(y)) \right] \\
&\leq |r(x) - r(y)| + \gamma \mathbb{E}_{X' \sim P^{\pi_e}(\cdot|x), Y' \sim P^{\pi_e}(\cdot|y)}[d_m(X', Y')] \\
&\quad + \frac{1}{2} \mathbb{E}_{\substack{X_1', X_2' \sim P^{\pi_e}(\cdot|x) \\ Y_1', Y_2' \sim P^{\pi_e}(\cdot|y)}}[|r(X_1') - r(X_2')| + |r(Y_1') - r(Y_2')|] \\
&\quad + \frac{1}{2} \sum_{n=1}^{m+1} \gamma^n (\Delta_n^{\pi_e}(x) + \Delta_n^{\pi_e}(y)) \\
&= |r(x) - r(y)| + \gamma \mathbb{E}_{X' \sim P^{\pi_e}(\cdot|x), Y' \sim P^{\pi_e}(\cdot|y)}[d_m(X', Y')] + \frac{1}{2} \sum_{n=0}^{m+1} \gamma^n (\Delta_n^{\pi_e}(x) + \Delta_n^{\pi_e}(y)) \\
&= d_{m+1}(x, y) + \frac{1}{2} \sum_{n=0}^{m+1} \gamma^n (\Delta_n^{\pi_e}(x) + \Delta_n^{\pi_e}(y))
\end{aligned}
$$

We thus have $|q_{m+1}(x) - q_{m+1}(y)| \leq d_{m+1}(x, y) + \frac{1}{2} \sum_{n=0}^{m+1} \gamma^n (\Delta_n^{\pi_e}(x) + \Delta_n^{\pi_e}(y))$, which completes the proof. $\square$

Lemma 3 tells us that the KROPE state-actions that are close in latent space also have similar action-values upto a constant $C := \frac{1}{2} \sum_{n=0}^{m+1} \gamma^n (\Delta_n^{\pi_e}(x) + \Delta_n^{\pi_e}(y))$. Intuitively, $\Delta_n^{\pi_e}(x)$ is the expected absolute reward difference between two trajectories at the $n$th step after $\pi_e$ is rolled out from $x$. If the transition dynamics and $\pi_e$ are deterministic, we have $C = 0$ (Castro, 2020; Zhang et al., 2021). Note that while the deterministic transition dynamics assumption is eliminated, the bound suggests that KROPE may hurt accuracy of $\hat{q}^{\pi_e}$ since when $d_{\text{KROPE}}(x, y) = 0$, we get $|q^{\pi_e}(x) - q^{\pi_e}(y)| \leq C$. This indicates that two state-actions that may have different action-values are considered the same under KROPE. This implies that while $x$ and $y$ should have different representations, they actually may have the same representation.

## B.2. KROPE Stability

In this section we present our main results. We present supporting theoretical results in Section B.2.1 and main theoretical results in Section B.3. To the best of our knowledge, even the supporting proofs in Section B.2.1 are new.

### B.2.1. SUPPORTING THEORETICAL RESULTS

We present the following definitions that we refer to in our proofs.

**Definition 8** (Bellman completeness (Chen & Jiang, 2019)). *The function class $\mathcal{F}$ is said to be Bellman complete if $\forall f \in \mathcal{F}$, it holds that $\mathcal{T}^{\pi_e} f \in \mathcal{F}$. That is $\sup_{f \in \mathcal{F}} \inf_{g \in \mathcal{F}} \|g - \mathcal{T}^{\pi_e} f\|_\infty = 0$, where $\mathcal{F} \subset \mathcal{X} \to [\frac{r_{min}}{1-\gamma}, \frac{r_{max}}{1-\gamma}]$, and $\mathcal{T}^{\pi_e}$ is the Bellman operator.*

**Definition 9** (Piece-wise constant functions (Chen & Jiang, 2019)). *Given a state-action abstraction $\phi$, let $\mathcal{F}^\phi \subset \mathcal{X} \to [\frac{r_{min}}{1-\gamma}, \frac{r_{max}}{1-\gamma}]$. Then $f \in \mathcal{F}^\phi$ is said to be a piece-wise constant function if $\forall x, y \in \mathcal{X}$ where $\phi(x) = \phi(y)$, we have $f(x) = f(y)$.*

**Proposition 4.** *If a state-action abstraction function $\phi : \mathcal{X} \to \mathcal{X}^\phi$ is a $\pi_e$-bisimulation abstraction, then $\mathcal{F}^\phi$ is Bellman complete, that is, $\sup_{f \in \mathcal{F}^\phi} \inf_{f' \in \mathcal{F}^\phi} \|f' - \mathcal{T}^{\pi_e} f\|_\infty = 0$.*

*Proof.* We first define $\pi_e$-bisimulation (Castro, 2020). Note that (Castro, 2020) considered only state abstractions, while we consider state-action abstractions. $\phi$ is considered a $\pi_e$-bisimulation abstraction if it induces a mapping between $\mathcal{X}$ and $\mathcal{X}^\phi$ such that for any $x, y \in \mathcal{X}$ such that $x, y \in \phi(x)$, we have:

1. $r(x) = r(y)$

2. $\forall x^\phi \in \mathcal{X}^\phi, \sum_{x' \in x^\phi} P^{\pi_e}(x'|x) = \sum_{x' \in x^\phi} P^{\pi_e}(x'|y)$

Given our $\pi_e$-bisimulation abstraction function $\phi$, we can group state-actions actions according to its definition above. Once we have this grouping, according to Definition 9, $\phi$ induces a piece-wise constant (PWC) function class $\mathcal{F}^\phi$. Note that by definition of $\phi$ we have:

$$\epsilon_r := \max_{x_1, x_2 : \phi(x_1) = \phi(x_2)} |r(x_1) - r(x_2)| = 0$$

$$\epsilon_p := \max_{x_1, x_2 : \phi(x_1) = \phi(x_2)} \left| \sum_{x' \in x^\phi} P^{\pi_e}(x'|x_1) - \sum_{x' \in x^\phi} P^{\pi_e}(x'|x_2) \right| = 0, \forall x^\phi \in \mathcal{X}^\phi.$$

Once we have $\phi$, we consider the following to show Bellman completeness. Our proof closely follows the proof of Proposition 20 from (Chen & Jiang, 2019). First recall the definition of Bellman completeness from Definition 8: $\forall f \in \mathcal{F}, \forall \mathcal{T}^{\pi_e} f \in \mathcal{G}$, $\sup_{f \in \mathcal{F}} \inf_{g \in \mathcal{G}} \|g - \mathcal{T}^{\pi_e} f\|_\infty = 0$. Given that the smallest value $\forall f \in \mathcal{F}, \forall \mathcal{T}^{\pi_e} f \in \mathcal{G}$, $\sup_{f \in \mathcal{F}} \inf_{g \in \mathcal{G}} \|g - \mathcal{T}^{\pi_e} f\|_\infty$ can take on is zero, we will prove our claim by showing that $\forall f \in \mathcal{F}, \forall \mathcal{T}^{\pi_e} f \in \mathcal{G}$, $\sup_{f \in \mathcal{F}} \inf_{g \in \mathcal{G}} \|g - \mathcal{T}^{\pi_e} f\|_\infty$ is upper-bounded by zero when $\phi$ is a $\pi_e$-bisimulation.

We will prove the upper bound by showing that there exists a function $f' \in \mathcal{F}^\phi$ such that $\|f' - \mathcal{T}^{\pi_e} f\|_\infty \leq 0$, which implies that $\inf_{f' \in \mathcal{F}^\phi} \|f' - \mathcal{T}^{\pi_e} f\|_\infty \leq 0$.

We now construct such a $f' \in \mathcal{F}^\phi$. We first define the following terms for a given abstract state-action $x^\phi \in \mathcal{X}^\phi$: $x_+ := \arg\max_{x \in \phi^{-1}(x^\phi)} (\mathcal{T}^{\pi_e} f)(x)$ and $x_- := \arg\min_{x \in \phi^{-1}(x^\phi)} (\mathcal{T}^{\pi_e} f)(x)$. We can then define $f'$ as follows:

$$f'(x) := \frac{1}{2}((\mathcal{T}^{\pi_e} f)(x_+) + (\mathcal{T}^{\pi_e} f)(x_-)), \forall x \in x^\phi.$$

And since this holds true for $\forall x \in x^\phi$, $f'_1$ is piece-wise constant function. We can then upper bound $\|f' - \mathcal{T}^{\pi_e} f\|_\infty$ as

follows:

$$f_1'(x) - (\mathcal{T}^{\pi_e} f)(x)$$

$$\leq \frac{1}{2}((\mathcal{T}^{\pi_e} f)(x_+) + (\mathcal{T}^{\pi_e} f)(x_-)) - (\mathcal{T}^{\pi_e} f)(x_-)$$

$$= \frac{1}{2}((\mathcal{T}^{\pi_e} f)(x_+) - (\mathcal{T}^{\pi_e} f)(x_-))$$

$$= \frac{1}{2}(r(x_+) + \gamma \mathbb{E}_{x_+' \sim P^{\pi_e}(x_+)}[f^{\pi_e}(x_+')] - r(x_-) - \gamma \mathbb{E}_{x_-' \sim P^{\pi_e}(x_-)}[f^{\pi_e}(x_-')])$$

$$\leq \frac{\gamma}{2} \left| \mathbb{E}_{x_+' \sim P^{\pi_e}(x_+)}[f^{\pi_e}(x_+')] - \mathbb{E}_{x_-' \sim P^{\pi_e}(x_-)}[f^{\pi_e}(x_-')] \right| \tag{1}$$

$$= \frac{\gamma}{2} \left| \sum_{x' \in \mathcal{X}} [f^{\pi_e}(x')(P^{\pi_e}(x'|x_+) - P^{\pi_e}(x'|x_-))] \right|$$

$$= \frac{\gamma}{2} \left| \sum_{x^\phi \in \mathcal{X}^\phi} \left( \sum_{x' \in x^\phi} f^{\pi_e}(x') P^{\pi_e}(x'|x_+) - \sum_{x' \in x^\phi} f^{\pi_e}(x') P^{\pi_e}(x'|x_-) \right) \right|$$

$$= \frac{\gamma}{2} \left| \sum_{x^\phi \in \mathcal{X}^\phi} f^{\pi_e}(x^\phi) \left( \sum_{x' \in x^\phi} P^{\pi_e}(x'|x_+) - \sum_{x' \in x^\phi} P^{\pi_e}(x'|x_-) \right) \right| \tag{2}$$

$$= \frac{\gamma}{2} \left| \sum_{x^\phi \in \mathcal{X}^\phi} f^{\pi_e}(x^\phi) \left( \Pr(x^\phi|x_+) - \Pr(x^\phi|x_-) \right) \right| \tag{3}$$

$$\leq \frac{\gamma}{2} \left\| \Pr(x^\phi|x_+) - \Pr(x^\phi|x_-) \right\|_1 \cdot \|f^{\pi_e}(x^\phi)\|_\infty \tag{4}$$

$$\leq 0 \qquad\qquad\qquad \epsilon_p = 0$$

where $\Pr$ denotes probability, (1) is due to $\max_{x_1, x_2 : \phi(x_1) = \phi(x_2)} |r(x_1) - r(x_2)| = 0$, (2) is due to $f^{\pi_e}(x^\phi) = f^{\pi_e}(x), \forall x \in x^\phi$ since PWC, (3) is due to $\Pr(x^\phi|x) = \sum_{x' \in x^\phi} P^{\pi_e}(x'|x)$, and (4) is due to Hölder's, $\|f(g)g(x)\|_1 \leq \|f(x)\|_1 \|g(x)\|_\infty$.

Similarly, we can show the other way around: $(\mathcal{T}^{\pi_e} f)(x) - f_1'(x) \leq 0$ by giving the symmetric argument starting with $(\mathcal{T}^{\pi_e} f)(x) - f_1'(x) \leq (\mathcal{T}^{\pi_e} f)(x_+) - \frac{1}{2}((\mathcal{T}^{\pi_e} f)(x_+) + (\mathcal{T}^{\pi_e} f)(x_-))$. Therefore, when $\phi$ is a $\pi_e$-bisimulation, we have $\sup_{f \in \mathcal{F}^\phi} \inf_{f' \mathcal{F}^\phi} \|f' - \mathcal{T}^{\pi_e} f\|_\infty = 0$. $\qquad\square$

**Lemma 4.** *Define the matrix $K_1 \in \mathbb{R}^{|\mathcal{X}| \times |\mathcal{X}|}$ such that each entry is the short-term similarity, $k_1$, between every pair of state-actions, i.e., $K_1(s_1, a_1; s_2, a_2) := 1 - \frac{1}{|r_{max} - r_{min}|} |r(s_1, a_1) - r(s_2, a_2)|$. Then $K_1$ is a positive semidefinite matrix.*

*Proof.* Proposition 2.21 from (Paulsen & Raghupathi, 2016) states that any kernel $k$ is positive semidefinite if it takes the form: $k(a, b) = \min\{a, b\}$ where $a, b \in [0, \infty)$.

First, recall that $r(s, a) \in [-1, 1]$, we then have each entry in the $K_1$ matrix of the following kernel form $K_1(x, y) = 1 - \frac{1}{2}|x - y|$. We can then re-write $k_1$ as follows:

$$k_1(x, y) = 1 - \frac{1}{2}|x - y|$$

$$= 1 + \frac{1}{2} \min\{-x, -y\} + \frac{1}{2} \min\{x, y\}$$

$$= \underbrace{\frac{1}{2} \min\{1 - x, 1 - y\}}_{k_a} + \underbrace{\frac{1}{2} \min\{1 + x, 1 + y\}}_{k_b}.$$

That is,

$$k_1(x, y) = k_a(x, y) + k_b(x, y).$$

Since $x \in [-1, 1]$, each term in the $\min$ function is non-negative. Thus, $k_a$ and $k_b$ are both positive semidefinite kernels, which means $k_1$ is also a positive semidefinite kernel. We then have that $K_1$ is a positive semidefinite matrix. $\qquad\square$

**Lemma 5.** *Given a finite set $\mathcal{X}$ and a kernel $k$ defined on $\mathcal{X}$, let $K = (k(x,y))_{x,y \in \mathcal{X}} \in \mathbb{R}^{|\mathcal{X}| \times |\mathcal{X}|}$ be the corresponding kernel matrix. If $K$ is full-rank and $MMD(k)(p,q) = 0$ for two probability distributions $p$ and $q$ on $\mathcal{X}$, then $p = q$.*

*Proof.* From (Gretton et al., 2012), we have the definition of MMD between two probability distributions $p, q$ given kernel $k$:

$$\text{MMD}(k)(p,q) := \|\mathbb{E}_{x \sim p}[k(x, \cdot)] - \mathbb{E}_{x \sim q}[k(x, \cdot)]\|_{\mathcal{H}_k}.$$

Now when $\text{MMD}(k)(p,q) = 0$, we have:

$$0 = \|\mathbb{E}_{x \sim p}[k(x, \cdot)] - \mathbb{E}_{x \sim q}[k(x, \cdot)]\|_{\mathcal{H}_k},$$

which implies

$$0 = \|\mathbb{E}_{x \sim p}[k(x, \cdot)] - \mathbb{E}_{x \sim q}[k(x, \cdot)]\|_2$$

since all norms are equivalent in a finite-dimensional Hilbert space. With $p$ and $q$ viewed as vectors in $\mathbb{R}^{|\mathcal{X}|}$, the above equality means

$$0 = \|Kp - Kq\|_2.$$

Hence, $K(p - q) = 0$. Since $K$ is full rank by assumption, we conclude that $p = q$. $\qquad\square$

**Lemma 6.** *Suppose we have a reproducing kernel $k$ defined on the finite space $\mathcal{X}$, which produces a reproducing kernel Hilbert space (RKHS) $\mathcal{H}_k$, with the induced distance function $d$ such that $d(x,y) = k(x,x) + k(y,y) - 2k(x,y), \forall x, y \in \mathcal{X}$. When $d(x,y) = 0$, then $k(x, \cdot) = k(y, \cdot)$.*

*Proof.* When $d(x,y) = 0$, we have $2k(x,y) = k(x,x) + k(y,y)$. Therefore, we the following equalities:

$$
\begin{aligned}
k(x,x) + k(y,y) &= 2k(x,y) \\
k(x,x) - k(x,y) &= k(x,y) - k(y,y) \\
\langle k(x,\cdot), k(x,\cdot) \rangle_{\mathcal{H}_k} - \langle k(x,\cdot), k(y,\cdot) \rangle_{\mathcal{H}_k} &= \langle k(x,\cdot), k(y,\cdot) \rangle_{\mathcal{H}_k} - \langle k(y,\cdot), k(y,\cdot) \rangle_{\mathcal{H}_k} \qquad (1) \\
\langle k(x,\cdot), k(x,\cdot) - k(y,\cdot) \rangle_{\mathcal{H}_k} &= \langle k(x,\cdot) - k(y,\cdot), k(y,\cdot) \rangle_{\mathcal{H}_k} \qquad (2) \\
\langle k(x,\cdot), k(x,\cdot) - k(y,\cdot) \rangle_{\mathcal{H}_k} &= \langle k(y,\cdot), k(x,\cdot) - k(y,\cdot) \rangle_{\mathcal{H}_k} \qquad (3) \\
\implies k(x,\cdot) &= k(y,\cdot)
\end{aligned}
$$

where (1), (2), and (3) are is due to RKHS definition, linearity of inner product, and symmetry of inner product respectively. $\qquad\square$

**Proposition 5.** *Let $x_1, \ldots, x_n \in (0, \infty)$ be $n$ distinct and strictly positive numbers. Let $K \in \mathbb{R}^{n \times n}$ be the matrix with entries $K_{ij} = \min\{x_i, x_j\}$. Then $K$ is a positive definite matrix.*

*Proof.* By Proposition 2.21 in (Paulsen & Raghupathi, 2016), the matrix $K$ is positive semidefinite, so we only need to show that $K$ is full rank. WLOG assume that $0 < x_1 < x_2 < \cdots < x_n$. We prove by induction on $n$. The base case with $n = 1$ clearly holds. Suppose the claim holds for $n - 1$ numbers. Now consider $n$ numbers. Let $\alpha = (\alpha_1, \ldots, \alpha_n)^\top \in \mathbb{R}^n$. It suffices to show that $K\alpha = 0$ implies $\alpha = 0$. We write $K$ in block matrix form as

$$
K = x_1 J_n + \begin{bmatrix} 0 & 0 & \cdots & 0 \\ 0 & x_2 - x_1 & \cdots & x_2 - x_1 \\ \vdots & \cdots & \ddots & \cdots \\ 0 & x_2 - x_1 & \cdots & x_n - x_1 \end{bmatrix} = x_1 \begin{bmatrix} 1 & 1 & \cdots & 1 \\ \mathbf{1} & \mathbf{1} & \cdots & \mathbf{1} \end{bmatrix} + \begin{bmatrix} 0 & 0 & \cdots & 0 \\ \mathbf{0} & \mathbf{u}_2 & \cdots & \mathbf{u}_n \end{bmatrix},
$$

where $J_n$ is the $n$-by-$n$ all one matrix, $\mathbf{1} \in \mathbb{R}^{n-1}$ the all one vector, $\mathbf{0} \in \mathbb{R}^{n-1}$ the all zero vector, and $\mathbf{u}_i \in \mathbb{R}^{n-1}, i = 2, \ldots, n$. It follow that

$$
0 = K\alpha = \begin{bmatrix} x_1 \sum_{i=1}^n \alpha_i \\ (x_1 \sum_{i=1}^n \alpha_i) \mathbf{1} + \sum_{i=2}^n \alpha_i \mathbf{u_i} \end{bmatrix},
$$

that is,

$$x_1 \sum_{i=1}^{n} \alpha_i = 0, \tag{7}$$

$$\left( x_1 \sum_{i=1}^{n} \alpha_i \right) \mathbf{1} + \sum_{i=2}^{n} \alpha_i \mathbf{u_i} = \mathbf{0}. \tag{8}$$

Plugging equation (7) into equation (8), we get $\sum_{i=2}^{n} \alpha_i \mathbf{u_i} = \mathbf{0}$. By the induction hypothesis, the $(n-1)$-by-$(n-1)$ matrix

$$\begin{bmatrix} \mathbf{u_2} & \cdots & \mathbf{u_n} \end{bmatrix} = \begin{bmatrix} \min\{x_i - x_1, x_j - x_1\} \end{bmatrix}_{i,j=2,\dots,n}$$

has full rank since the $(n-1)$ numbers $x_2 - x_1, \dots, x_n - x_1$ are distinct and strictly positive. Therefore, we must have $\alpha_2 = \cdots = \alpha_n = 0$. Plugging back into equation (7) and using $x_1 > 0$, we obtain $\alpha_1 = 0$. □

## B.3. Main KROPE Theoretical Results

We now present the main theoretical contributions of our work.

**Theorem 1.** *If $\Phi$ is a KROPE representation as defined in Definition 2, then the spectral radius of $(\mathbb{E}[\Phi^{\top}\Phi]))^{-1}\mathbb{E}[\gamma\Phi^{\top}P^{\pi_e}\Phi]$ is less than 1. That is, $\Phi$ stabilizes LSPE.*

*Proof.* Recall from Definition 2, we have:

$$\mathbb{E}[\Phi\Phi^{\top}] = K_1 + \gamma\mathbb{E}[P^{\pi_e}\Phi(P^{\pi_e}\Phi)^{\top}],$$

where $K_1 \in \mathbb{R}^{|\mathcal{X}| \times |\mathcal{X}|}$ such that each entry is the short-term similarity, $k_1$, between every pair of state-actions i.e. $K_1(s_1, a_1; s_2, a_2) := 1 - \frac{|r(s_1,a_1) - r(s_2,a_2)|}{|r_{\max} - r_{\min}|}$.

From this definition, we can proceed by left and right multiplying $\Phi^{\top}$ and $\Phi$ respectively to get:

$$\mathbb{E}[\Phi^{\top}\Phi\Phi^{\top}\Phi] = \mathbb{E}[\Phi^{\top}K_1\Phi] + \gamma\mathbb{E}[\Phi^{\top}P^{\pi_e}\Phi(P^{\pi_e}\Phi)^{\top}\Phi].$$

Notice that $B := \mathbb{E}[\Phi^{\top}\Phi]$ is the feature covariance matrix and $C := \mathbb{E}[\Phi^{\top}P^{\pi_e}\Phi]$ is the cross-covariance matrix. By making the appropriate substitutions, we get:

$$BB^{\top} = \mathbb{E}[\Phi^{\top}K_1\Phi] + \gamma CC^{\top}.$$

We can then left and right multiply by $B^{-1}$ and $B^{-\top}$ to get the following where $L := \gamma B^{-1}C$:

$$I = B^{-1}\mathbb{E}[\Phi^{\top}K_1\Phi]B^{-\top} + \frac{1}{\gamma}LL^{\top}.$$

Rearranging terms gives

$$I - \frac{1}{\gamma}LL^{\top} = B^{-1}\mathbb{E}[\Phi^{\top}K_1\Phi]B^{-\top}.$$

From Lemma 4, we know that $K_1$ is positive semidefinite, which means that $B^{-1}\mathbb{E}[\Phi^{\top}K_1\Phi]B^{-\top}$ is also positive semidefinite. Therefore, the eigenvalues of LHS above must also be greater than or equal to zero. Letting $\lambda$ be the eigenvalue of $L$, we know that that the following must hold:

$$1 - \frac{\lambda^2}{\gamma} \geq 0 \implies |\lambda| \leq \sqrt{\gamma}.$$

Since $\gamma < 1$, the spectral radius of $L = (\mathbb{E}[\Phi^{\top}\Phi])^{-1}(\gamma\mathbb{E}[\Phi^{\top}P^{\pi_e}\Phi])$ is always less than 1. Thus, KROPE representations are stable. Finally, since KROPE representations are stable and due to Proposition 1, KROPE representations stabilize LSPE. □

**Theorem 2.** *Let $\phi : \mathcal{X} \to \mathcal{X}^{\phi}$ be the state-action abstraction induced by grouping state-actions $x, y \in \mathcal{X}$ such that if $d_{\text{KROPE}}(x, y) = 0$, then $\phi(x) = \phi(y), \forall x, y \in \mathcal{X}$. Then $\phi$ is Bellman complete if the abstract reward function $r^{\phi} : \mathcal{X}^{\phi} \rightarrowtail (-1, 1)$ is injective (i.e., distinct abstract rewards).*

*Proof.* Our proof strategy is to show that the abstraction function $\phi$ due to KROPE is a $\pi_e$-bisimulation, which implies it is Bellman complete due to Proposition 4.

According to Propsition 3, $d_{\text{KROPE}}(x, y) = |r(x) - r(y)| + \gamma\text{MMD}(k^{\pi_e})(P^{\pi_e}(\cdot|x), P^{\pi_e}(\cdot|y))$. When $d_{\text{KROPE}}(x, y) = 0$ for any two state-actions, it implies that $r(x) = r(y)$ and $\text{MMD}(k^{\pi_e})(P^{\pi_e}(\cdot|x), P^{\pi_e}(\cdot|y)) = 0$.

For $\phi$ to be a $\pi_e$-bisimulation, we need $\forall x^\phi \in \mathcal{X}^\phi, \sum_{x' \in x^\phi} P^{\pi_e}(x'|x) = \sum_{x' \in x^\phi} P^{\pi_e}(x'|y)$ to be true for any $x, y \in \mathcal{X}$ such that $\phi(x) = \phi(y)$. While $\text{MMD}(k^{\pi_e})(P^{\pi_e}(\cdot|x), P^{\pi_e}(\cdot|y)) = 0$, it is possible that $P^{\pi_e}(\cdot|x) \neq P^{\pi_e}(\cdot|y)$. However, as we will show, under the assumption that the abstract rewards $r^\phi$ are distinct $\forall x^\phi \in \mathcal{X}^\phi$, we do have $\forall x^\phi \in \mathcal{X}^\phi, \sum_{x' \in x^\phi} P^{\pi_e}(x'|x) = \sum_{x' \in x^\phi} P^{\pi_e}(x'|y)$. Before we proceed, we make the following technical assumption on the reward function: $r(x) \in (-1, 1), \forall x \in \mathcal{X}$. The exclusion of the rewards $-1$ and $1$ allows us to use Proposition 5 to show that the KROPE kernel is positive definite instead of positive semi-definite.

Once we group state-actions $x, y \in \mathcal{X}$ together such that $d_{\text{KROPE}}(x, y) = 0$, we have the corresponding abstraction function $\phi : \mathcal{X} \to \mathcal{X}^\phi$. Accordingly, $\phi$ induces a Markov reward process, $\mathcal{M}^\phi := \langle \mathcal{X}^\phi, r^\phi, P^\phi, \gamma \rangle$ where $r^\phi$ is the abstract reward function $r^\phi : \mathcal{X}^\phi \to (-1, 1)$ and $P^\phi$ is the transition dynamics on the abstract MRP i.e. $P^\phi(\cdot|x^\phi)$. We can also consider the abstract KROPE kernel, $k^\phi(x^\phi, y^\phi)$, which measures the KROPE relation on $\mathcal{X}^\phi$. Note that all these quantities are a function of $\pi_e$. We drop the notation for clarity. By this construction, we have:

$$r^\phi(x^\phi) = r(x), \forall x \in x^\phi \qquad\qquad \text{Since all rewards are equal within } x^\phi$$

$$k^\phi(x^\phi, \cdot) = k(x, \cdot), \forall x \in x^\phi \qquad\qquad \text{Lemma 6}$$

Now, under the assumption that all abstract rewards $r^\phi(x^\phi)$ are distinct $\forall x^\phi \in \mathcal{X}^\phi$, we have that the kernel matrix $K^\phi \in \mathbb{R}^{\mathcal{X}^\phi \times \mathcal{X}^\phi}$ where each entry $k^\phi(x^\phi, y^\phi)$ is positive definite. To see this fact, consider that:

$$k^\phi(x^\phi, y^\phi) = k_1^\phi(x^\phi, y^\phi) + \gamma\mathbb{E}_{X^\phi \sim P^\phi(\cdot|x^\phi), Y^\phi \sim P^\phi(\cdot|y^\phi)}[k^\phi(X^\phi, Y^\phi)], \qquad (9)$$

where $k_1^\phi(x^\phi, y^\phi) := 1 - \frac{1}{r_{\max}^\phi - r_{\min}^\phi}|r^\phi(x^\phi) - r^\phi(y^\phi)|$. From Lemma 4, we know that $k_1^\phi$ is positive semidefinite. However, under the assumption that all abstract rewards $r^\phi$ are distinct, Proposition 5 tells us that $k_1^\phi$ is positive definite. Given that $k^\phi(x^\phi, y^\phi)$ (Equation (9)) is just a summation of positive definite kernels, $k^\phi$ is positive definite, which means $K^\phi$ is positive definite.

We now consider when the MMD is zero. Again, by construction, we have the following when $\text{MMD}(k^{\pi_e})(P^{\pi_e}(\cdot|x), P^{\pi_e}(\cdot|y)) = 0$. For clarity, we use $k$ instead of $k^{\pi_e}$.

$$\begin{aligned}
0 &= \|\mathbb{E}_{X' \sim P^{\pi_e}(\cdot|x)}[k(X', \cdot)] - \mathbb{E}_{X' \sim P^{\pi_e}(\cdot|y)}[k(X', \cdot)]\|_{\mathcal{H}_k} \\
&= \|\mathbb{E}_{X' \sim P^{\pi_e}(\cdot|x)}[k(X', \cdot)] - \mathbb{E}_{X' \sim P^{\pi_e}(\cdot|y)}[k(X', \cdot)]\|_2 \\
&= \|\sum_{x' \in \mathcal{X}} P^{\pi_e}(x'|x)k(x', \cdot) - \sum_{x' \in \mathcal{X}} P^{\pi_e}(x'|y)k(x', \cdot)\|_2 \\
&= \left\|\sum_{x^\phi \in \mathcal{X}^\phi}\sum_{x' \in x^\phi} P^{\pi_e}(x'|x)k(x', \cdot) - \sum_{x^\phi \in \mathcal{X}^\phi}\sum_{x' \in x^\phi} P^{\pi_e}(x'|y)k(x', \cdot)\right\|_2 \\
&= \left\|\sum_{x^\phi \in \mathcal{X}^\phi} k^\phi(x^\phi, \cdot)\sum_{x' \in x^\phi} P^{\pi_e}(x'|x) - \sum_{x^\phi \in \mathcal{X}^\phi} k^\phi(x^\phi, \cdot)\sum_{x' \in x^\phi} P^{\pi_e}(x'|y)\right\|_2 \qquad (1) \\
&= \left\|\sum_{x^\phi \in \mathcal{X}^\phi} k^\phi(x^\phi, \cdot)\Pr(x^\phi|x) - \sum_{x^\phi \in \mathcal{X}^\phi} k^\phi(x^\phi, \cdot)\Pr(x^\phi|y)\right\|_2 \qquad \text{Pr denotes probability}
\end{aligned}$$

where (1) is due to $k^\phi(x^\phi, \cdot) = k(x, \cdot), \forall x \in x^\phi$ From above, we can see that the kernel and probability distributions are over $\mathcal{X}^\phi$. In matrix notation, we can write the above as follows where $p^\phi := \Pr(\cdot|x)$ and $q^\phi := \Pr(\cdot|y)$ are viewed as

probability distribution vectors in $\mathbb{R}^{|\mathcal{X}^\phi|}$.

$$0 = \left\| K^\phi p^\phi - K^\phi q^\phi \right\|_2$$
$$\implies p^\phi = q^\phi \qquad \text{since } K^\phi \text{ is positive definite, from Lemma 5.}$$

We thus have $\forall x^\phi \in \mathcal{X}^\phi, \sum_{x' \in x^\phi} P^{\pi_e}(x'|x) = \sum_{x' \in x^\phi} P^{\pi_e}(x'|y)$ to be true for any $x, y \in \mathcal{X}$ such that $\phi(x) = \phi(y)$. Given this condition holds true and $r(x) = r(y), \forall x, y \in \mathcal{X}$ such that $\phi(x) = \phi(y)$, $\phi$ is a $\pi_e$-bisimulation. From Proposition 4 we then have that $\phi$ is Bellman complete.

$\square$

**Remark on the injective reward assumption**. The injective reward assumption simply means that each abstract state-action group will have a distinct associated reward from every other abstract state-action group. This assumption comes as a trade-off. Chen & Jiang (2019) proved that bisimulation abstractions are Bellman complete. Instead of assuming injective rewards, they assumed that two states were grouped together if each state's transition dynamics led to next states that are also grouped together (one of the conditions for exact bisimulations). This condition is also considered strict and inefficient to compute (Castro, 2020).

In our work, we relax this exact transition dynamics equality by considering independent couplings between next state distributions, thereby making the KROPE algorithm efficient to compute. However, the drawback of this relaxation is that preserving distinctness between abstract state-actions may be lost (see remarks on $k^{\pi_e}$ in Section 3.1). To ensure the distinctness between state-action abstractions, we assumed that the reward function is injective. This then allowed us to show Bellman completeness in a similar way to that shown in Chen & Jiang (2019).

## C. Empirical Details

In this section, we provide specific details on the empirical setup and additional results.

### C.1. Empirical Setup

**General Training Details.** In all the continuous state-action experiments, we use a neural network with 1 layer and 1024 neurons using RELU activation function and layernorm to represent the encoder $\phi : \mathcal{X} \to \mathbb{R}^d$ (Gallici et al., 2025). We use mini-batch gradient descent to train the network with mini-batch sizes of 2048 and for 500 epochs, where a single epoch is a pass over the full dataset. We use the Adam optimizer with learning rate $\{1e^{-5}, 2e^{-5}, 5e^{-5}\}$ and weight decay $1e^{-2}$. The target network is updated with a hard update after every epoch. The output dimension $d$ is $\{|\mathcal{X}|/4, |\mathcal{X}|/2, 3|\mathcal{X}|/4\}$, where $|\mathcal{X}|$ is the dimension of the original state-action space of the environment. All our results involve analyzing this learned $\phi$. Since FQE outputs a scalar, we add a linear layer on top of the $d$-dimensional vector to output a scalar. The entire network is then trained end-to-end. The discount factor is $\gamma = 0.99$. The auxiliary task weight with FQE for all representation learning algorithms is $\alpha = 0.1$. When using LSPE for OPE, we invert the covariance matrix by computing the pseudoinverse.

In the tabular environments, we use a similar setup as above. The only changes are that we use a linear network with a bias component but no activation function and fix the learning rate to be $1e^{-3}$. For the experiment in Section 4.4, $\alpha = 0.8$. We refer the reader pseudo-code in Appendix A.

**Evaluation Protocol: OPE Error**. As noted earlier, we measure OPE error by measuring MSVE. To ensure comparable and interpretable values, we normalize the MSVE by dividing with $\text{MSVE}[q^{\text{RAND}}] := \mathbb{E}_{(S,A) \sim \mathcal{D}}[(q^{\text{RAND}}(S, A) - q^{\pi_e}(S, A))^2]$, where $q^{\text{RAND}}$ is the action-value function of a random-policy. Similarly, in the continuous state-action environments, we normalize by $\text{MSVE}[q^{\text{RAND}}] := \mathbb{E}_{S_0 \sim d_0, A_0 \sim \pi_e}[(q^{\text{RAND}}(S_0, A_0) - q^{\pi_e}(S_0, A_0))^2]$. Values less than one mean that the algorithm estimates the true performance of $\pi_e$ better than a random policy.

**Evaluation Protocol: Realizability Error**. In tabular experiments, we normalize the realizability error. After solving the least-squares problem $\epsilon := \|\Phi\hat{w} - q^{\pi_e}\|_2^2$, where $\hat{w} := \arg\min_w \|\Phi w - q^{\pi_e}\|_2^2$. We divide $\epsilon$ by $\frac{1}{|\mathcal{X}|} \sum_i |q^{\pi_e}(x_i)|$ and plot this value.

**Pearson Correlation.** The formula for the Pearson correlation used in the main experiments is:

$$r = \frac{\sum_{i=1}^{n} (x_i - \bar{x})(y_i - \bar{y})}{\sqrt{\sum_{i=1}^{n} (x_i - \bar{x})^2} \sqrt{\sum_{i=1}^{n} (y_i - \bar{y})^2}}$$

where $\bar{x}$ and $\bar{y}$ are the means of all the $x_i$'s and $y_i$'s respectively.

**Custom Datasets.** We generated the datasets by first training policies in the environment using SAC (Haarnoja et al., 2018) and recording the trained policies during the course of training. For each environment, we select 3 policies, where each contributes equally to generate a given dataset. We set $\pi_e$ to be one of these policies. The expected discounted return of the policies and datasets for each domain is given in Table 2 ($\gamma = 0.99$). In all environments, $\pi_e = \pi_b^1$ (see Table 2). The values for the evaluation and behavior policies were computed by running each for 300 unbiased Monte Carlo rollouts, which was more than a sufficient amount for the estimate to converge. This process results in total of 4 datasets, each of which consisted of 100K transitions.

| Environments | $\pi_e$ | $\pi_b^1$ | $\pi_b^2$ |
|---|---|---|---|
| CartPoleSwingUp | 50 | 20 | 5 |
| FingerEasy | 100 | 71 | 32 |
| HalfCheetah | 51 | 27 | 2 |
| WalkerStand | 90 | 55 | 40 |

*Table 2.* Policy values of the target policy and behavior policy on DM-control (Tassa et al., 2018).

**D4RL Datasets.** Due to known discrepancy issues between newer environments of gym[2], we generat our datasets instead of using the publicly available ones. To generate the datasets, we use the publicly available policies [3]. For each domain, the expert (and target policy) was the 10th (last policy) from training. The medium (and behavior policy) was the 5th policy. We added a noise of 0.1 to the policies. The values for the evaluation and behavior policies were computed by running each for 300 unbiased Monte Carlo rollouts , which was more than a sufficient amount for the estimate to converge. We set $\gamma = 0.99$. We evaluate on the Cheetah, Walker, and Hopper domains. This generation process for three environments, led to 9 datasets, each of which consisted of 100K transitions.

**Toy Divergence Example** We set $\gamma = 1$. The sampled state-actions are fed into a linear network encoder with a bias component and no activation function which outputs a $d = 3$ representation. This representation is shaped by KROPE and this representation is then fed into a linear function to output the scalar value for FQE. $\mathcal{D}^{\text{on}}$ is a dataset of 2000 on-policy transitions. $\mathcal{D}^s$ is an off-policy dataset with 5000 transitions starting from the specified state $s$, where $s = \{w_1, w_2, w_3, 2w_3\}$, where the next state is sampled according to the transition dynamics of the MRP.

Regarding the statement of, "we would generally expect that the probability of sampling a bad transition *pair* is less than that of sampling a single bad transition", we make the following simplistic and rough calculation. For a given a dataset, assume that the probability of choosing a bad transition is $p$ and a good transition is $1 - p$. Then sampling a pair of bad transitions versus sampling a single bad transition involves comparing the probabilities of $p^2$ and $2p(1 - p)$, respectively. In general, we would expect $2p(1 - p) \geq p^2$, unless $p \geq 0.66$, i.e., the dataset starts to overflow with bad transitions. Roughly, we would expect FQE to diverge when $p \geq 0.5$, but we would expect KROPE to diverge when $p \geq 0.66$, which suggests that KROPE is less prone to divergence than FQE.

### C.2. Baselines

We provide details of the 6 baselines in this section.

**DBC.** It is a bisimulation-based metric learning algorithm that learns $\pi$-bisimilar representations by ensuring the L1 distance between the latent representations approximates the $\pi$-bisimulation metric (Zhang et al., 2021). During the learning

---

[2]https://github.com/Farama-Foundation/D4RL/tree/master
[3]https://github.com/google-research/deep_ope

process, it also learns a model of the environment. Its additional hyperparameters are the learning rate of model learning (which we set to $1e^{-5}$) and the output dimension of $\phi$.

**ROPE.** It is a bisimulation-based metric learning algorithm that is off-policy evaluation variant of MICO (Castro et al., 2021; Pavse & Hanna, 2023a). It learns representations directly such that the $\pi$-bisimulation metric is satisfied and makes no assumptions on the transition dynamics. Its additional learning rate is the output dimension of $\phi$. Furthermore, we provide additional details on our experimental setup is different from that in (Pavse & Hanna, 2023a). The main differences are:

1. The original ROPE used ROPE as a pre-training step, fixed the representations, and then fed them into FQE for OPE. In our case, we also pre-train the representations but with FQE as a representation learning algorithm (for value predictive representations (Lehnert & Littman, 2020)) along with other representation learning algorithms as auxiliary tasks. The fixed learned representations are then fed into LSPE for OPE.

2. The original ROPE encoder architecture had a TANH activation function on the output layer, which we effectively serves a clipping mechanism to the features, which is similar to how public implementations of FQE clip the return to avoid divergence[4].

We opted to use our alternative setup for two reasons: 1) By using LSPE as the OPE algorithm, instead of FQE, we can precisely quantify the stability properties of the representations in terms of the spectral radius (Theorem 1), which is harder to do when using FQE as the OPE algorithm; 2) while a valid architectural choice, for this current work, we viewed the use of the tanh function as obfuscating the true stability properties of the representations and so opted to avoid it. More practically, it is reasonable to use the TANH as part of the architecture.

**BCRL.** Unlike the other algorithms, BCRL is not used as an auxiliary loss with FQE (Chang et al., 2022). We use the same learning rates as mentioned above for $\{1e^{-5}, 2e^{-5}, 5e^{-5}\}$ when training $\phi$. As suggested by prior work, self-predictive algorithms such BCRL work well when network that outputs the predicted next state-action is trained at a faster rate (Tang et al., 2023; Grill et al., 2020). Accordingly, we set its learning rate to be $1e^{-4}$. For BCRL-EXP, which involves the log determinant regularizer, we set this coefficient to $1e^{-2}$. BCRL's hyperparameters are: the learning rates for $\phi$, $M$, and $\rho$; the output dimension of $\phi$; and the log determinant coefficient (see Equation (10)).

**DR3.** The DR3 regularizer minimizes the total feature co-adaptation by adding the term $\sum_{(s,a,s')\in\mathcal{D},a'\sim\pi_e}\phi(s,a)^\top\phi(s',a')$ as an auxiliary task to the main FQE loss (Kumar et al., 2022). (Ma et al., 2024) introduced an improvement to this auxiliary loss by suggesting that the absolute value of the feature co-adaptation be minimized, i.e., $\sum_{(s,a,s')\in\mathcal{D},a'\sim\pi_e}|\phi(s,a)^\top\phi(s',a')|$. We use $\alpha = 0.1$ as its auxiliary task weight. Absolute DR3's only hyperparameters are the auxiliary task weight $\alpha$ and the $\phi$ output dimension.

**BEER.** (He et al., 2024) introduced an alternative regularizer to DR3 rank regularizer since they suggested that the minimization of the unbounded feature co-adaptation can undermine performance. They introduced their bounded rank regularizer BEER (see Equation (12) in (He et al., 2024)). BEER introduces only the auxiliary task weight $\alpha$ as the additional hyperparameter.

**KROPE.** KROPE's only hyperparameters are the output dimension of $\phi$ and the learning rate of the KROPE learning algorithm.

### C.3. Additional Results

In this section, we include additional empirical results.

**Realizability.** In addition to stability, we also care about the realizability of $\Phi$. We say $\Phi$ is a *realizable* representation if $q^{\pi_e} \in \text{Span}(\Phi)$, where $\text{Span}(\Phi)$ is the subspace of all expressible action-value functions with $\Phi$. Note that even if $\Phi$ is a realizable and stable representation, LSPE may not recover the $q^{\pi_e}$ solution (Sutton & Barto, 2018).

A basic criterion for learning $q^{\pi_e}$ is realizability. That is, we want $\epsilon := \|\Phi\hat{w} - q^{\pi_e}\|_2^2$, where $\hat{w} := \arg\min_w \|\Phi w - q^{\pi_e}\|_2^2$, to be low. In our experiments, we compute $\epsilon$ and plot it as a function of $d$ in Figure 5(a). A critical message from our

---

[4]https://github.com/google-research/google-research/blob/master/policy_eval/q_fitter.py

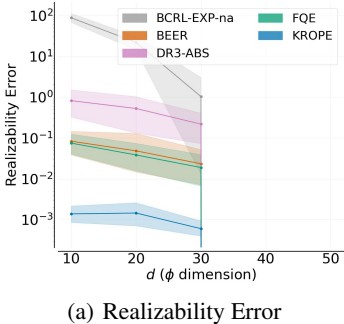

(a) Realizability Error

*Figure 5.* Evaluation of basic representation properties on Garnet MDPs with 40 state-actions vs. output dimension $d$. Figure 5(a): Realizability error; lower is better. All results are averaged over 30 trials and the shaded region is the 95% confidence interval.

results is that stability and realizability do not always go hand-in-hand. While BCRL-EXP-NA has favorable spectral radius properties (Figure 2(a)), it has poor realizability, which will negatively affect its OPE accuracy. KROPE, on the other hand, has favorable stability and realizability properties up till $d = 30$. When $d \geq 40$, the realizability error is 0 for all algorithms since the subspace spanned by $\Phi$ is large enough to contain the true action-value function (Ghosh & Bellemare, 2020). While the realizability error is 0 for $d \geq 40$, the representations can be unstable (Figure 2(a)).

### C.3.1. LEARNING GRAPHS FOR OFFLINE POLICY EVALUATION ON DMC AND D4RL DATASETS

In this section, we present the offline policy evaluation results on the DMC and D4RL datasets. We present the results in Figures 6 and 7.

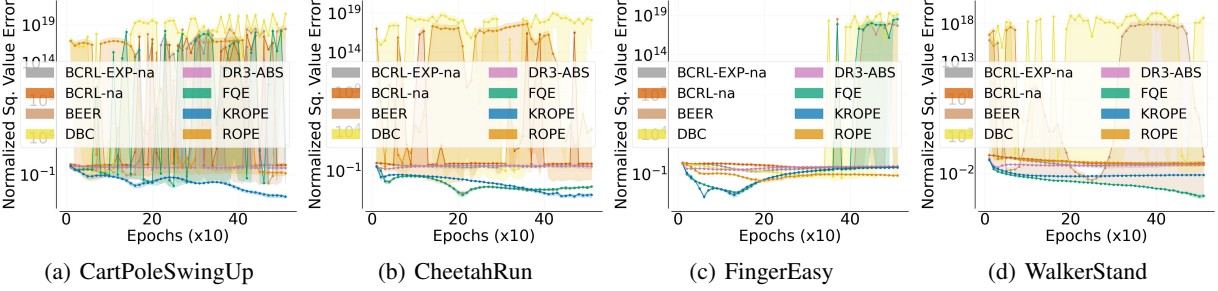

| (a) CartPoleSwingUp | (b) CheetahRun | (c) FingerEasy | (d) WalkerStand |

*Figure 6.* Normalized squared value error achieved by LSPE when using a particular representation vs. representation training epochs on the custom DMC datasets. LSPE estimates are computed every 10 epochs. Results are averaged over 20 trials and the shaded region is the 95% confidence interval. Lower and less erratic is better.

On the DMC dataset, we find that KROPE reliably produces stable and low MSVE during the full course of training.

On the D4RL dataset, we reach the similar conclusions: KROPE is effective in producing stable and accurate OPE estimates. However, in $3/9$ instances, KROPE does diverge. This divergence is likely related to the discussion in Section 5 and Section 4.4. Recall that KROPE is a semi-gradient method, which does not optimize any objective function and is susceptible to divergence (Feng et al., 2019; Sutton & Barto, 2018). So while KROPE *representations* stabilize value function learning, KROPE's *learning algorithm* may diverge and not converge to KROPE representations. However, regardless of this result, KROPE does improve the stability and accuracy of FQE in all cases.

### C.3.2. STABILITY-RELATED ANALYSIS ON CUSTOM DATASETS

In this section, We include the remaining stability-related metric analysis that was deferred from the main paper.

**Hyperparameter Sensitivity.** In this subsection, we include all the remaining results related to the stability metrics for all environments.

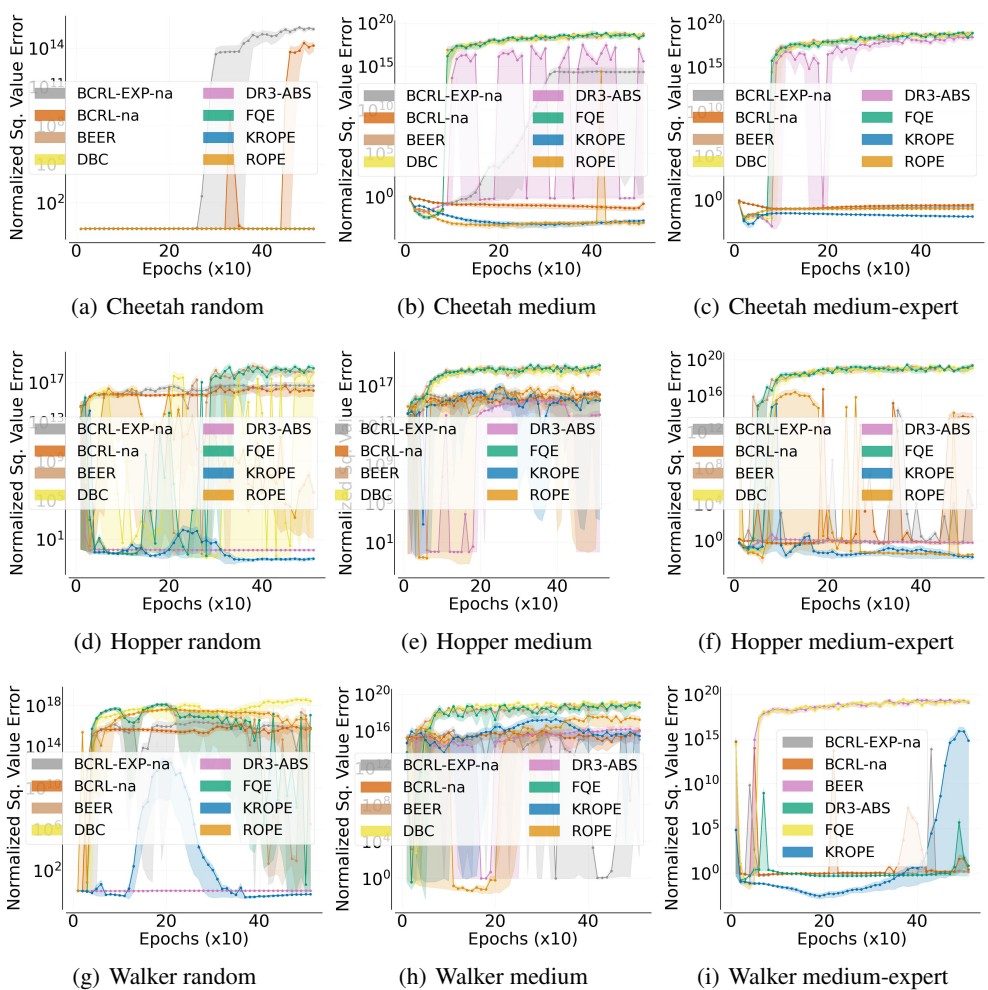

*Figure 7.* Normalized squared value error achieved by LSPE when using a particular representation vs. representation training epochs on the D4RL datasets. LSPE estimates are computed every 10 epochs. Results are averaged over 20 trials and the shaded region is the 95% confidence interval. Lower and less erratic is better.

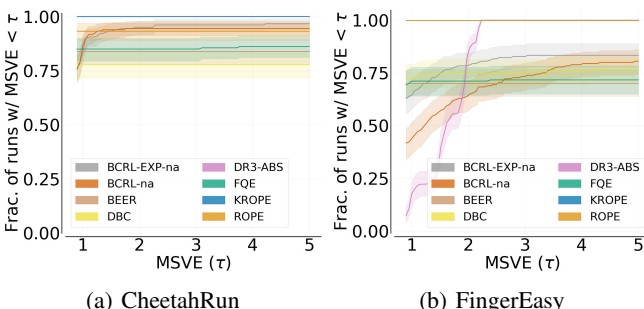

*Figure 8.* Hyperparameter sensitivity on different environments as a function of training epochs; larger area under the curve is better. All results are averaged over 20 trials for each hyperparameter configuration and shaded region is the 95% confidence interval.

**Bellman completeness.** Another metric that is associated with stability is Bellman completeness (BC) (Chang et al., 2022; Wang et al., 2021). We find that KROPE is approximately Bellman complete even though it does not explicitly optimize for it; this finding aligns with our Theorem 2. While BC is difficult to approximate, we can minimize the proxy metric introduced given in Equation (10) (Chang et al., 2022):

$$\mathcal{L}(M, \rho) := \mathbb{E}_{\mathcal{D}} \left\| \begin{bmatrix} M \\ \rho^\top \end{bmatrix} \phi(s, a) - \begin{bmatrix} \gamma \mathbb{E}_{s' \sim P(\cdot|s,a), a' \sim \pi_e(\cdot|s')}[\phi(s', a')] \\ r(s, a) \end{bmatrix} \right\|_2^2 \tag{10}$$

where $(\rho, M) \in \mathbb{R}^{d \times d}$, $\phi$ is fixed, and $\mathcal{L}(M, \rho) = 0$ indicates Bellman completeness. Given the final learned representation, we compute and report the BC error in Table 3. We find that KROPE is approximately Bellman complete even though it does not explicitly optimize for it; this finding aligns with our Theorem 2. We note that BCRL-EXP is less Bellman complete since it also includes the exploratory objective in its loss function, which if maximized can reduce the Bellman completeness. While BCRL is more BC than BCRL-EXP, we found that it is less BC in general. We attribute this finding due to the difficulty in explicitly optimizing the BCRL objective which involves multiple neural networks ($M, \rho, \phi$) and multiple loss functions on different scales (reward, self-prediction, log determinant regularization losses). KROPE can achieve approximate Bellman completeness without these optimization-related difficulties.

| Domain | BCRL + EXP | BCRL | BEER | DR3 | FQE | KROPE (ours) |
|---|---|---|---|---|---|---|
| | | | Algorithm | | | |
| CartPoleSwingUp | $0.4 \pm 0.1$ | $0.2 \pm 0.1$ | $0.1 \pm 0.0$ | $0.0 \pm 0.0$ | $0.1 \pm 0.0$ | $0.0 \pm 0.0$ |
| CheetahRun | $3.3 \pm 0.6$ | $2.4 \pm 0.5$ | $0.7 \pm 0.0$ | $0.0 \pm 0.0$ | $0.7 \pm 0.0$ | $0.2 \pm 0.0$ |
| FingerEasy | $1.3 \pm 0.6$ | $0.7 \pm 0.2$ | $0.9 \pm 0.0$ | $137.0 \pm 4.4$ | $0.9 \pm 0.0$ | $0.2 \pm 0.0$ |
| WalkerStand | $10.4 \pm 2.0$ | $0.3 \pm 0.1$ | $0.5 \pm 0.1$ | $66.1 \pm 0.6$ | $0.6 \pm 0.0$ | $0.1 \pm 0.0$ |

*Table 3.* Bellman completeness measure for all algorithms across all domains. Results are averaged across 20 trials and the deviation shown is the 95% confidence interval. Values are rounded to the nearest single decimal.

### C.3.3. USING FQE DIRECTLY FOR OPE

In our main empirical section (Section 4), we used FQE as a representation learning algorithm on our custom datasets. We adopted the linear evaluation protocol, i.e., an approach of analyzing the penultimate features of the action-value function network and applied LSPE on top of these features for OPE. This protocol enabled us to better understand the nature of the learned features.

For the sake of completeness, we present results of FQE as an OPE algorithm where the action-value network is directly used to estimate the performance of $\pi_e$. We present the results in Figures 9 and 10. As done in Section 4, we evaluate the performance of FQE and KROPE based on how they shape the penultimate features of the action-value network. However, when conducting OPE, we evaluate two variants: 1) using LSPE (-L) and 2) using the same end-to-end FQE action-value network (-E2E).

From Figure 9, we find that there are hyperparameter configurations that can outperform the KROPE variants. However, based on Figure 10, we find both KROPE variants are significantly more robust to hyperparameter tuning. This latter result suggests that KROPE does improve stability with respect to the hyperparameter sensitivity metric as well.

Regardless of FQE's hyperparameter sensitivity, it is still interesting to observe that when FQE is used as an OPE algorithm, it produces reasonably accurate OPE estimates. It even outperforms the FQE+KROPE combination. The primary difference between using FQE for OPE vs. FQE features and LSPE for OPE is in how the last linear layer is trained. The former is trained by gradient descent while the latter is trained with the iterative LSPE algorithm on the fixed features. An interesting future direction will be to explore the learning dynamics of these two approaches.

On a related note, we point out that the training dynamics of FQE are still not well-understood. For example, (Fujimoto et al., 2022) show that the FQE loss function poorly correlates with value error. That is, the FQE loss can be high but value error (and OPE error) can be low.

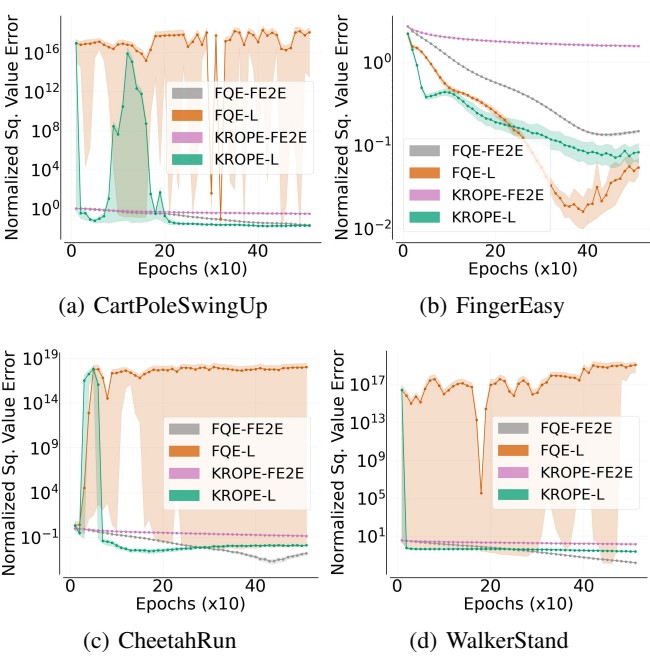

*Figure 9.* Normalized squared value error achieved by LSPE (-L) and FQE (-E2E) evaluated every 10 epochs of training. Results are averaged over 20 trials and the shaded region is the 95% confidence interval. Lower and less erratic is better.

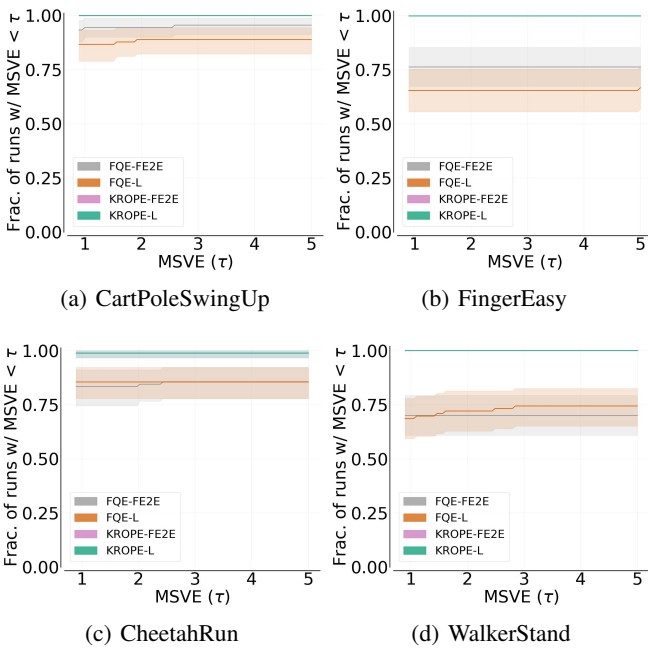

*Figure 10.* Hyperparameter sensitivity on different environments as a function of training epochs; larger area under the curve is better. All results are averaged over 10 trials for each hyperparameter configuration and shaded region is the 95% confidence interval. We tuned the hyperparameters discussed in Appendix C.1. KROPE-FE2E overlaps with KROPE-L.

