# OpenReview forum: "Stable Offline Value Function Learning with Bisimulation-based Representations"
_ICML.cc/2025/Conference — ICML 2025 poster_

### Official Review · Reviewer_KFHs · 2025-03-12

**Overall Recommendation:** 3

**Summary:**

The paper tackles the field of offline policy evaluation (OPE) and addresses methodology to find good state-action-pair representations. It introduces a kernel based state-action representation and gives theoretical properties for it. It then presents experimental results of the introduced KROPE method on different benchmarks.

## update after rebuttal
Score increased. See rebuttal comment.

**Claims And Evidence:**

- The statement in the abstract "Therefore, it is critical to stabilize value function learning by explicitly shaping the state-action representations." is not supported by the experiments presented. It is shown in the paper that for some cases value function learning was successful without learning any state-action representation.

- The statement "In this work, we investigate how to explicitly learn state-action representations to stabilize value function learning." is misleading. The work introduces one way to do so, but does not investigate different approaches in my understanding.

- There is a list of 5 contributions under the headline "Can bisimulation-based representation learning stabilize offline value function learning?". The fourth and fifth claim are slightly misleading. The fifth claim can not be part of the main paper since its contents are inside the appendix. The claim that KROPE representations can be successfully used in OPE is shown in the paper. To the best of my understanding the evidence is not convincing that KROPE representations always lead to more stable and accurate offline value function learning.

- The Takeaway #1 is not true in general and thus, it needs clarification.

- Takeaway #2 is not true in general. At least from my understanding this is not given in general.

**Essential References Not Discussed:**

The reference to Van Hasselt et al., 2018 that coined the term "deadly triad" which is used throughout the paper is missing.

**Experimental Designs Or Analyses:**

Table 1 is not informative. This needs thorough revision. Stating numbers in a table is a good idea in general, but the mean +/- the standard error should be the standard to do so.

**Methods And Evaluation Criteria:**

The proposed methods and benchmarks seem reasonable for the scope of the work.

**Other Comments Or Suggestions:**

- Paragraph title "Remarks..." in 190 is weirdly formatted, as well as subsection title of 4.2
- fitted *Q*-evaluation should have at least an uppercased Q

**Other Strengths And Weaknesses:**

- The paper lacks clarity in some parts of the submission.

- Formatting is not ICML compliant, e.g., the uppercased abbreviations.

- General formatting needs streamlining.

- In 209 right side it says that continuity is important. But why?

- The ylabels in several plots are very hard to read and leave room for improvement.

- Overall this work is an interesting read with valuable contents in need of a major revision and re-evalution of the made claims.

**Questions For Authors:**

- Can you elaborate on my concerns regarding the claims made in the paper?

- In the experiments in the appendix, you contradict the statement that KROPE always leads to more stable results, or do I missinterpretate the presented results?

**Relation To Broader Scientific Literature:**

The paper contributes to the OPE literature.

**Theoretical Claims:**

- All theoretical claims are based on the standard coverage assumption that gives a non-zero probability of state-action pairs appearing in the offline dataset for finite state spaces.
- The LSPE stabilization looks fine to me.
- The Bellman completeness proof looks fine as well.

---

> ### Author Rebuttal · Authors · 2025-03-29
>
> Thank you very much for your thoughtful comments and feedback. Thank you for acknowledging that the work was an interesting read with valuable contents.
>
> Your comments are helpful in making our paper precise. We do believe, however, that these adjustments involve minor reframings/edits. We address your concerns below.
>
> **Claims and evidence**
>
> All the concerns in this subsection are easily addressable since we acknowledge the shortcomings of KROPE explicitly in Section 5 and Appendix C.3.1. All changes are sentence-level clarifications and involve bringing discussions from Section 5 earlier in the paper.
>
> **Re: “’Therefore, it is critical to stabilize value function learning by explicitly shaping the state-action representations.’ is not supported by the experiments presented.”**
>
> That statement is a general desired property we want from representation learning algorithms for offline policy evaluation. With regards to KROPE, we already discuss KROPE’s inability to stabilize in all settings in Section 5 and Appendix C.3.1. That said, we understand the concern and will discuss the limitation earlier in the paper.
>
> **Re: “"In this work, we investigate how to explicitly learn state-action representations to stabilize value function learning." is misleading.”**
>
> The statement means we show how one may go about learning KROPE representations. But we understand that this can be potentially confusing. We do not investigate different approaches to shape the representations. We propose only one way to do so, and we will clarify that.
>
> **Re: “The fourth and fifth claim are slightly misleading."**
>
> We will adjust contribution 4 based on our discussion in Section 5 and Appendix C.3.1 to state that it does indeed improve the stability of OPE compared to 7 other baselines on 10/13 datasets (note: 10 does not include just highlighted blue errors in table 1). Regarding Contribution 5, we will try to move it to the main paper.
>
> **Re: “The Takeaway #1 is not true in general and thus, it needs clarification.”**
>
> We understand the point, we will rephrase the takeaway to explicitly say that the statement is true under theoretical assumptions.
>
> **Re: “Takeaway #2 is not true in general. At least from my understanding this is not given in general.”**
>
> Takeaway #2 is true in an aggregate sense across datasets (10/13) and compared to the 7 other baselines. But we understand the point. We will state that KROPE improves the chances of stability for OPE compared to other representation learning baselines.
>
> The more accurate characterization of our work is the following: *our theoretical results prove that if state-action features satisfy the KROPE relation (Definition 2), then they will lead to stable value function learning. Practically, since KROPE relies on a semi-gradient method (like DQN/fitted Q-evaluation; see Section 5 and Appendix C.3.1), the algorithm may still lead to divergence. Empirically, KROPE improves stability compared 7 other baselines, and leads to stable and accurate OPE in 10/13 cases (note: 10 does not include just highlighted blue errors in table 1). Therefore, KROPE improves upon other baselines in learning stable and accurate representations for OPE.*
>
> We will accordingly discuss this earlier in the paper instead of waiting till Section 5 and modify the claims/takeways.
>
> **Re: “Table 1 is not informative. This needs thorough revision. Stating numbers in a table is a good idea in general, but the mean +/- the standard error should be the standard to do so.”**
>
> Thanks for your feedback. However, the table does include information for statistical rigor. Each value is the MSE over 20 trials and the range shows the 95% confidence interval, which are all important measures of statistical rigor (see caption of Table 1) [1]. We also include learning curves in Figures 6/7 (appendix).
>
> **Deadly triad reference**
>
> Thanks for the reference. We will include this.
>
> **Formatting issues: ICML compliant, ylabels, line 190 format, FQE upper case**
>
> Thanks and we will update this and the graphs to be clearer.
>
> **Meaning of continuity**
>
> We will add clarification on this and we refer the reviewer to the work of Le Lan et al. [2] for continuity of metrics in the context of RL (see their Figure 1 in Le Lan [2]). Briefly, ideally, states that have similar values are close to each other in feature space. This statement is true for general machine learning too: inputs with similar outputs should ideally be close to each other in feature space.
>
> Once again thank you for making our work more precise. We believe the main concern on claims is easily addressable. Please let us know if we have addressed your concerns. If we have, we would greatly appreciate it if you could re-evaluate your review.
>
> ---
> [1] Patterson et al. 2024. Empirical Design in Reinforcement Learning.
>
> [2] Le Lan et al. 2021. Metrics and continuity in reinforcement learning.

---

> > ### Comment · Reviewer_KFHs · 2025-04-07
> >
> > Thank you for your comprehensive rebuttal and for addressing the points raised in my initial review.
> > After reading the rebuttal and the other reviews, I am considering raising my score from 2 to 3.

---

> > > ### Author Response · Authors · 2025-04-07
> > >
> > > Thank you so much for raising the score and for your response! We appreciate your effort in making the paper stronger. When you get a chance, please do update the score in the main review. We would greatly appreciate it. Thanks again!

---

### Official Review · Reviewer_YH4r · 2025-03-12

**Overall Recommendation:** 4

**Summary:**

This paper introduces Kernel Representations for Offline Policy Evaluation (KROPE), a kernel-based representation learning algorithm based on bisimulation metric-like ideas. They study a class of representations which emerge as the solution to the representation learning loss, and prove that it has desirable theoretical properties, in particular being stable for off-policy value function learning and is Bellman-complete under additional assumptions. They then evaluate across a range of both tabular and larger-scale environments, and both validate their theoretical results and compare KROPE across a range of baselines for OPE.

**Claims And Evidence:**

All claims made in the submission are supported by proper evidence.

**Essential References Not Discussed:**

I don't believe any essential references are not discussed.

**Experimental Designs Or Analyses:**

The experimental design and validity seems good to me (I only very quickly skimmed the code provided).

**Methods And Evaluation Criteria:**

The methods and evaluation done make sense for the problem at hand.

**Other Comments Or Suggestions:**

- I quite like the introduction and discussion of the KROPE representation $\Phi$ in Definition 2. To reassure the reader, can you add perhaps a minor which states that if $\Phi$ is a KROPE representation, then the inner product between two state-action pairs is equal to the kernel evaluated at them?
- It could strengthen the paper to shorten sections 1 & 2 and move some of Appendix C.3. to the main text. There is quite a bit of background/discussion before the novel contributions begin (halfway through page 4), and the main takeaways should be in the main body (takeaway #3 is currently in the appendix).
- In section C.2., in the description of ROPE, minor nit: "Its additional learning rate is the output dimension of $\phi$."

**Other Strengths And Weaknesses:**

**Strengths**
- The application of stability analysis to bisimilarity metric-type representations is a nice perspective and I expect it to be built upon in the future. Additionally the proof of Theorem 1 appears rather general (I think it should apply with at least any reasonable choice of immediate similarity kernel).
- The paper is well-written and easy to follow.
- The choice of experiments in Section 4.2. nicely complement the theoretical results of section 3.

**Weaknesses**
- Theorem 2 is dependent on a very strong assumption (that the reward function $r^\phi$ is injective), without any discussion around the assumption or what it may entail. I think that this assumption is violated in almost any non-contrived setting I can think of (from large-scale complex ones like Atari/MuJoCo to gridworlds, maze-based, CartPole, etc), which I believe limits the impact of the result.

**Questions For Authors:**

- I'm a bit confused by Definition 2 -- in particular what does $\mathbb{E}_{\mathcal{D}} [\Phi\Phi^T]$ represent for a matrix $\Phi \in \mathbb{R}^{|\mathcal{X}| \times d}$ (what depends on $\mathcal{D}$)?
- Similar to my first comment in the previous section -- is it possible for there to be a feature mapping $\Phi$ such that $\langle \phi(x,a), \phi(y,b)\rangle = k^{\pi_e}(x,a;y,b)$ but $\Phi$ is not a KROPE representation?
- Can there be a weaker statement similar to Theorem 2, without assuming the fact that $r^\phi$ is injective (likely the statement would depend on $r^\phi$, and the current theorem might appear as a special case).

**Relation To Broader Scientific Literature:**

In a sense the key contributions of this paper can be seen as extending the analysis/understanding of bisimulation-based representation learning to stability-type results, which I think is an important contribution on its own, and I expect it to lead to further research.

**Theoretical Claims:**

I checked all proofs, and did not find any issues.

---

> ### Author Rebuttal · Authors · 2025-03-29
>
> Thank you very much for appreciating our work and the clarity of our writing. Thank you for mentioning the strengths of our empirical and theoretical work, especially Theorem 1 and our choice of experiments. We also appreciate your acknowledging the significance of our results and potential for further research
>
> **Theorem 2 is dependent on a very strong assumption (that the reward function is injective)**
>
> The injective reward assumption simply means that each abstract state-action group will have a distinct associated reward from every other abstract state-action group. While we agree that it is strong, it comes as a tradeoff. Chen and Jiang [1] proved that bisimulation abstractions are Bellman complete. Instead of assuming injective rewards, they assumed that two states were grouped together if each state’s transition dynamics led to next states that are also grouped together (one of the conditions for exact bisimulations). This condition is also considered strict and inefficient to compute [2].
>
> In our work, we relax this exact transition dynamics equality by considering independent couplings between next state distributions, thereby making the KROPE algorithm efficient to compute. However, the drawback is that preserving distinctness between abstract state-actions may be lost (see page 4 Section 3.1 on remarks on $k^{\pi_e}$). To ensure the distinctness between state-action abstractions, we assumed that the reward function is injective. This then allowed us to show Bellman completeness in a similar way to that shown in Chen and Jiang [1].
>
> Regarding a weaker version: it may be possible to relax this assumption and consider the error induced in Bellman completeness. Ultimately, what is needed is the ability to preserve distinctiveness between state-action features/abstractions. Ensuring such a property with a weaker condition would be interesting to investigate.
>
> Note that this assumption is only for the BC proof. Theorem 1 does not make this assumption. We will include the rationale behind the injective reward assumption for Theorem 2 in the appendix.
>
> **Definition 2 clarification**
>
> Yes, we will add the point that the inner product equals the kernel evaluated for those features to the camera ready. Regarding your other question, each row in the $\Phi$ matrix corresponds to the feature vector for a state-action pair, so the dependence on $\mathcal{D}$ means that the state-actions (i.e., the features) are sampled according to their appearance in the batch of data.
>
> **Other inner products/feature maps for KROPE kernel**
>
> This is a really interesting question that is worth looking into. The features are shaped based on the PSD kernel used. Currently, Definition 2 defines a KROPE representation to be one that satisfies the relationship in Definition 2 in terms of linear dot products (which is a function of $k_1$, which is PSD (see Lemma 4 in Appendix)), which are easier to reason about. That said, Definition 2 can be broadened to just include short-term ($k_1$, which is PSD) and long-term similarity, where long-term similarity is determined by, say, a Gaussian kernel. This could also be a valid KROPE representation, but unclear if it will also be a stable representation as defined in our work (in terms of the spectral radius of the features).
>
> **Deferring background in Sections 1/2 to Appendix and moving Appendix C.3 up**
>
> We will definitely re-review Sections 1 and 2 and see what we can defer to the Appendix. We do believe that it is important to be complete, even if slightly redundant for readers who may already be familiar with the background knowledge. If possible, we will then move Appendix C.3 to the main text since we also agree that Appendix C.3 brings useful insights regarding the potential instability of KROPE.
>
> ---
> [1] Chen and Jiang. 2019. Information-Theoretic Considerations in Batch Reinforcement Learning
>
> [2] Castro. 2020. Scalable methods for computing state similarity in deterministic Markov Decision Processes

---

> > ### Comment · Reviewer_YH4r · 2025-04-03
> >
> > I thank the authors for their rebuttal, and I maintain my positive rating of the paper.

---

> > > ### Author Response · Authors · 2025-04-03
> > >
> > > We sincerely appreciate your appreciation of our paper, your feedback, and your responding to our rebuttal.

---

### Official Review · Reviewer_5dn2 · 2025-03-13

**Overall Recommendation:** 3

**Summary:**

The paper introduces Kernel Representations for Offline Policy Evaluation (KROPE), a novel algorithm designed to stabilize offline value function learning in reinforcement learning. KROPE leverages π-bisimulation to shape state-action representations, ensuring that similar state-action pairs are represented consistently. This approach enhances convergence and reliability. The authors provide theoretical foundations that demonstrate KROPE's stability through non-expansiveness and Bellman completeness. Empirical results indicate that KROPE outperforms other baselines in terms of stability and accuracy.

**Claims And Evidence:**

The claims made in the submission are supported by clear and convincing evidence.

**Essential References Not Discussed:**

Related works that are essential to understanding the key contributions of the paper are currently cited/discussed in the paper.

**Experimental Designs Or Analyses:**

We have confirmed the rationality of the experimental design in the revised paper. The experiment is mainly divided into two parts. The first part is to verify the stability and qπe consistency of the KROPE algorithm in the Garnet MDPs environment. The second part is to verify whether the KROPE algorithm can bring low and stable MSVE and whether the KROPE algorithm is sensitive to hyperparameters on 4 tasks in DMC and 9 datasets in D4RL.

**Methods And Evaluation Criteria:**

The proposed methods and evaluation criteria do make sense for the problem.

**Other Comments Or Suggestions:**

Overall, the paper is well-organized, with clear ideas and compelling theoretical and experimental evidence. However, the paper's starting point could be articulated more clearly. For instance, how do value networks further contribute to accurately predicting the expected discounted return from a given state?

**Other Strengths And Weaknesses:**

Strengths:
1.The author introduces a novel approach for evaluating the value function, with experimental results demonstrating its superior accuracy and stability in estimation compared to other baseline methods.
2.The theoretical proof of this paper is rigorous, and the proof ideas are also clear.
Weaknesses:
1.This paper demonstrates that the KROPE algorithm can learn a value network that accurately estimates the expected discounted return for a given state. However, it does not further elaborate on the role or potential applications of this learned value network.

**Questions For Authors:**

1.We aim to gain insights into the particular implementation of representing the (s, a) pair. Specifically, we are curious about whether states and actions should be concatenated prior to being fed into the network for representation, or if they should be processed by separate networks and concatenated subsequently.
2.Currently, this work has been focused on the D4RL dataset, which comprises physical states. We are interested in exploring the effectiveness of the method on the V-D4RL dataset, wherein states are depicted as images.
3.We are curious about the potential applications of value networks when they can accurately estimate the expected discounted return for each state. We believe that such an accurate value network could facilitate learning an effective policy. In this context, we hope the author can provide a comparison between the performance of the policy learned through KROPE and classical offline RL algorithms (such as TD3-BC [1], CQL [2], etc.). If the author can provide the above experimental results and demonstrate that the policy learned by the KROPE algorithm performs comparably to or better than classical offline RL algorithms, we will consider increasing the paper's score.
[1] Fujimoto, Scott, and Shixiang Shane Gu. "A minimalist approach to offline reinforcement learning." Advances in neural information processing systems 34 (2021): 20132-20145.
[2] Kumar, Aviral, et al. "Conservative q-learning for offline reinforcement learning." Advances in neural information processing systems 33 (2020): 1179-1191.

**Relation To Broader Scientific Literature:**

This paper focuses on learning a value network that accurately estimates the expected discounted return for each state. The author enhances the estimation accuracy by learning a more effective representation, setting this approach apart from previous studies like FQE [1]. However, the paper does not further elaborate on the advantages or potential applications of learning such a value network.
[1] Le, Hoang, Cameron Voloshin, and Yisong Yue. "Batch policy learning under constraints." International Conference on Machine Learning. PMLR, 2019.

**Theoretical Claims:**

Partially. We have carefully examined some theoretical proofs, including Lemma 1. However, due to the inconsistency between our research direction and other theoretical knowledge, we are unable to determine whether the theoretical proof is correct.

---

> ### Author Rebuttal · Authors · 2025-03-29
>
> We thank the reviewer for acknowledging that our empirical results and theoretical results are rigorous, and that the paper is clear and well-organized.
>
> Below we address your concerns.
>
> **Re: reason to learn the value network (“the paper does not further elaborate on the advantages or potential applications of learning such a value network.”)**
>
> In our context, our goal is to estimate the value of a policy. Our contribution is particularly relevant to the off-policy evaluation (OPE) literature [1, 4, 5]. In OPE, we want to use an offline dataset generated by different policies to estimate the performance of another target policy since deploying the target policy directly in the environment may be risky or costly. One way to evaluate the performance of this target policy is to compute its value function. Therefore, accurately estimating the value function for a target policy becomes important.
>
> OPE is particularly important in safety-critical tasks such as healthcare [2] or in situations where it may be monetarily costly to deploy a potentially poor performing policy such as in recommendation systems [3].
>
> **Re: state-action input (“we are curious about whether states and actions should be concatenated prior to being fed into the network for representation, or if they should be processed by separate networks and concatenated subsequently.”)**
>
> This is an interesting question. In our work, we concatenate the state-action pair and feed it directly into the network (i.e., we do not process them separately). This practice is fairly common (eg, [6, 7]). Further investigation, which is beyond the scope of this work, would be required to determine how this alternative approach performs for OPE.
>
> **Re: image states (“We are interested in exploring the effectiveness of the method on the V-D4RL dataset, wherein states are depicted as images”)**
>
> This is an interesting future direction. In this present version of KROPE, we focussed on illustrating that KROPE improved the stability of OPE theoretically and on D4RL and deepmind control suite environments (across 13 datasets). A next step would be to apply these ideas to visual domains.
>
> **Re: using KROPE for offline control (“could facilitate learning an effective policy” and “demonstrate that the policy learned by the KROPE algorithm”)**
>
> Thanks for this suggestion. We also expect KROPE to help in the offline control setting and it is an interesting direction to explore. In the current scope, however, we focussed on OPE. While control is interesting, studying OPE independently is important due to: 1) its practical significance in AI safety and building trust worthy RL agents and 2) prediction is a fundamental part of RL and is worth studying in isolation.
>
> Please let us know if we have addressed your concerns. If we have, we would greatly appreciate it if you could re-evaluate your review.
>
> ---
> [1] Voloshin et al. Empirical Study of Off-Policy Policy Evaluation for Reinforcement Learning. 2021
>
> [2] Gottesman et al. 2018. Evaluating Reinforcement Learning Algorithms in Observational Health Setting
>
> [3] Li et al. 2011. Unbiased offline evaluation of contextual-bandit- based news article recommendation algorithms
>
> [4] Fu et al. 2021. Benchmarks for Deep Off-Policy Evaluation
>
> [5] Uehera et al. 2022. A Review of Off-Policy Evaluation in Reinforcement Learning
>
> [6] Chang et al. 2022. Learning Bellman Complete Representations for Offline Policy Evaluation
>
> [7] Pavse et al. 2023. State-Action Similarity-Based Representations for Off-Policy Evaluation

---

> > ### Comment · Reviewer_5dn2 · 2025-04-08
> >
> > Thank you for your detailed response. It has addressed most of my concerns. I have decided to increase my score to 3.

---

> > > ### Author Response · Authors · 2025-04-08
> > >
> > > We are glad that we were able to address your concerns. Thank you for your response, increasing your score, and making our paper stronger! We will incorporate all the clarifications in the camera ready.

---

### Official Review · Reviewer_3GQD · 2025-03-13

**Overall Recommendation:** 3

**Summary:**

This paper addresses offline policy evaluation in offline RL, which involves estimating expected returns of state-action pairs under a fixed policy using offline datasets. Stability in this estimation process is critical for accurate evaluation. The authors propose KROPE, a new method combining bisimulation-based representation learning (ROPE) with kernel methods. KROPE constructs state-action representations such that pairs with similar immediate rewards and subsequent policy-induced states share similar representations. Experimental results suggest that KROPE enhances the stability of offline value function learning compared to baseline methods.

**Claims And Evidence:**

The paper provides robust theoretical and experimental support for its claims, detailed in Sections 3 and 4, respectively. The theoretical part are solid.

**Essential References Not Discussed:**

NA

**Experimental Designs Or Analyses:**

The presentation and clarity of results in Table 1 need substantial improvement for supporting the claims regarding KROPE's stability:
1. Numerical results are currently presented with only one decimal place, making it difficult to discern clear differences among methods, e.g., 0.0 vs 0.0 or 0.1 vs 1.0. Higher numerical precision is necessary for meaningful comparisons.
1. The absence of MAE results (as recommended by the DOPE benchmark) further complicates interpreting and validating the results.
1. Notably, ROPE appears to diverge in the current experiments, contradicting convergence reported in the original ROPE paper (Pavse & Hanna, 2023a). This discrepancy raises concerns regarding experimental consistency and soundness. The authors should clearly explain this difference.

Learning curves illustrating training stability should be included prominently in the main text, given the paper's central claim regarding stability.

Additionally, some learning curves presented in Figure 7 (Appendix) reveal that KROPE does not converge in certain environments (e.g., Walker and Hopper). Besides, the large squared value errors from certain baseline methods obscure meaningful comparisons with more stable methods. Addressing this visualization issue is recommended.


--------
Pavse & Hanna, 2023a. State-action similarity-based representations for off-policy evaluation. NeurIPS 2023.

**Methods And Evaluation Criteria:**

While fundamentally sound, the methodology could be strengthened by addressing the evaluation score choice. The experimental section primarily presents squared value error, whereas the DOPE benchmark suggests using MAE for such evaluations (Fu et al., 2021). An explanation for this deviation or inclusion of MAE results would enhance the paper's credibility and relevance.

----------
Fu et al., 2021.Benchmarks for Deep Off-Policy Evaluation. ICLR 2021

**Other Comments Or Suggestions:**

1. The addition of algorithm pseudocode specific for KROPE method would enhance clarity.
1. I would raise my score if the concerns related to the experimental results are thoroughly addressed.

**Other Strengths And Weaknesses:**

NA

**Questions For Authors:**

1. Could the authors clarify whether any experimental settings differ from those used in the original ROPE paper?

**Relation To Broader Scientific Literature:**

NA

**Theoretical Claims:**

The theoretical framework is well-constructed with no apparent flaws in the claims or proofs provided.

---

> ### Author Rebuttal · Authors · 2025-03-29
>
> Thank you for acknowledging that our empirical results and theoretical results are solid. Below we address the concerns you raise.
>
> **Use of Mean Absolute Error vs. MSE**
>
> This suggestion is valid since we understand the MAE may be more robust to outliers. However, it does not diminish the validity of the results since the MSE is a common metric used in the OPE literature, see this OPE benchmark paper [1]. The MSE is considered to be a valid metric due to its relation to the variance and bias of estimators (even though bias/variance may not be reported). We also refer the reviewer to many fundamental and prominent works that evaluate OPE algorithms based on the MSE [3,4,5,6,7].
>
> **Numerical precision in Table 1**
>
> After analyzing the raw data, we found that including additional precision did not alter the big picture. Moreover, we were tight on horizontal space, so we truncated it at 1 decimal.
>
> **Relations to Pavse et al. 2023**
>
> This is a great point, which we will make clear in the paper. Below we state why we opted to use this setup and then explain the differences. Briefly, the cited ROPE paper did: 1) use FQE as the OPE algorithm and 2) used a tanh activation function on the last layer of the encoder.
>
> Instead of adopting Pavse et al.’s approach, we opted to use our alternative setup for two reasons: 1) By using LSPE as the OPE algorithm, instead of FQE, we can precisely quantify the stability properties of the representations in terms of the spectral radius (Theorem 1 and Section 4.2), which is harder to do when using FQE as the OPE algorithm; 2) while a valid architectural choice, for this current work, we viewed the use of the tanh function as obfuscating the true stability properties of the representations and so opted to avoid it. More practically, it is reasonable to use the tanh as part of the architecture.
>
> In more detail, the main differences are:
> 1. The original ROPE paper used ROPE as a pre-training step, fixed the representations, and then fed them into FQE for OPE. In our case, we also pre-train the representations but with FQE as a representation learning algorithm (for value predictive representation [8]) along with other representation learning algorithms as auxiliary tasks. The fixed learned representations are then fed into LSPE for OPE.
> 2. The original ROPE encoder architecture (in the cited paper) had a tanh activation function on the output layer, which we effectively serves a clipping mechanism to the features [9], which is similar to how public implementations of FQE clip the return to avoid divergence [2].
>
> We will make these differences explicit in the Appendix in the camera ready where we discuss the different baselines
>
> **Including learning curves**
>
> Thanks for the suggestion. These are included in the Appendix (Figure 6 and 7) due to lack of space in the main paper.
>
> **KROPE does not always converge**
>
> Yes, it is more accurate to say that KROPE improves the stability of OPE compared to other baselines, and we will reframe the paper as such. KROPE may not converge since KROPE relies on a semi-gradient learning algorithm. We discuss this limitation in Section 5 and provide insight into when this might occur in Appendix C.3.1.
>
> Briefly, it may be possible to leverage the Legendre-Fenchel transformation and replace the fixed-point loss function of semi-gradient methods with an equivalent expression that avoids semi-gradient learning. However, a drawback with this approach is that the new learning objective is a minimax procedure, which can be challenging to optimize in practice [10].
>
> We will modify our claim to say that *KROPE improves the stability of OPE compared to 7 other baselines on 10/13 datasets.* (not just highlighted blue errors in table 1).
>
> **Pseudo-code**
>
> Appendix (page 15) includes one, but we will make it clearer.
>
> Please let us know if we have addressed your concerns. If we have, we would greatly appreciate it if you could re-evaluate your review.
>
> ---
> [1] Voloshin et al. Empirical Study of Off-Policy Policy Evaluation for Reinforcement Learning. 2021
> [2] https://github.com/google-research/google-research/blob/master/policy_eval/q_fitter.py#L101
> [3] Chaudhari et al. 2024. Abstract Reward Processes: Leveraging State Abstraction for Consistent Off-Policy Evaluation
> [4] Liu et al. 2018. Breaking the Curse of Horizon: Infinite-Horizon Off-Policy Estimation
> [5] Thomas et al. 2016. Data-Efficient Off-Policy Policy Evaluation for Reinforcement Learning
> [6] Hanna et al. 2019. Importance Sampling Policy Evaluation with an Estimated Behavior Policy
> [7] Sachdeva et al. 2023. Off-Policy Evaluation for Large Action Spaces via Policy Convolution
> [8] Lehnart and Littman. 2020. Successor features combine elements of model-free and model-based reinforcement learning
> [9] Bhatt et al. 2024. CrossQ: Batch Normalization in Deep Reinforcement Learning for Greater Sample Efficiency and Simplicity
> [10] Feng et al. 2019. A Kernel Loss for Solving the Bellman Equation

---

> > ### Comment · Reviewer_3GQD · 2025-04-08
> >
> > Thanks to the authors for the rebuttal. Most of my concerns are addressed. I’m raising my score to 3.

---

> > > ### Author Response · Authors · 2025-04-08
> > >
> > > We are glad that we were able to address your concerns. Thank you for your response, increasing your score, and making our paper stronger! We will incorporate all the clarifications in the camera ready.

---

### Decision · Program_Chairs · 2025-05-01

**Decision:**

Accept (poster)

**Comment:**

This paper proposes KROPE, a method for offline policy evaluation via improved learned representations, based on prior work on bisimulation metrics. The authors provide theoretical results demonstrating the increased stability for least-squares policy evaluation when integrated with KROPE, that KROPE's representations are Bellman complete, and empirically demonstrate the stability and performance of KROPE.

Although there were some initial reservations on the impact and clarity of the work, the authors were able to clarify all issues during the rebuttal. All reviewers are now supportive of acceptance, as am I, after having reviewed the discussion and paper itself.